## PROCEEDINGS A

differential equations, analysis, integral equations

smooth stable manifold, non-instantaneous, impulsive, exponential dichotomy

**Author for correspondence:**
Yonghui Xia
e-mail: xiadoc@163.com

# Smooth stable manifolds for the non-instantaneous impulsive equations with applications to Duffing oscillators

Weijie Lu[1], Manuel Pinto[2] and Yonghui Xia[1]

[1]College of Mathematics and Computer Science, Zhejiang Normal University, Jinhua 321004, People's Republic of China
[2]Departamento de Matemáticas, Universidad de Chile, Santiago, Chile

YX, 0000-0001-8918-3509

In this paper, we present a theory of smooth stable manifold for the non-instantaneous impulsive differential equations on the Banach space or Hilbert space. Assume that the non-instantaneous linear impulsive evolution differential equation admits a uniform exponential dichotomy, we give the conditions of the existence of the global and local stable manifolds. Furthermore, $C^k$-smoothness of the stable manifold is obtained, and the periodicity of the stable manifold is given. Finally, an application to nonlinear Duffing oscillators with non-instantaneous impulsive effects is given, to demonstrate the existence of stable manifold.

## 1. Introduction

### (a) History

The theory of the invariant manifold plays an important role in the study of the dynamic systems. It is a powerful tool in reduction of high-dimensional systems, linearization of dynamic systems and bifurcation of homoclinic or heteroclinic loops. It was ripe basically at the beginning of 1970s (see [1–4]). A foundation of the modern invariant manifold theory was given in the monograph (Hirsch *et al.* [5]). They summarized the classical invariant manifold theory concerning

stable or unstable manifolds, centre manifolds and centre-stable or centre-unstable manifolds; further they put forward the 'pseudo-hyperbolic' for mapping in infinite dimensional space and presented a finer structure involving strong stable or strong unstable manifolds. Now, this theory has been extended to deterministic dynamic systems of finite or infinite dimension (e.g. Henry [6], Carr [7], Chow & Lu [8,9], Bates & Jones [10], Bates *et al.* [11], Zhang *et al.* [12,13], Barreira & Valls [14–16], Caraballo *et al.* [17] and Shen *et al.* [18]). Among these works, there are two important methods: (i) one is Lyapunov–Perron's method (see [19,20]), which is based on the variation of the constant formula and the exponential dichotomy; (ii) the other is Hadamard's method (see [21]), employing the Hadamard's graph teansforms. Moreover, all these results require a spectral gap condition, it is decided by distribution of Lyapunov exponents and it is required to guarantee the smoothness of the invariant manifolds. In addition, using these methods, Foias *et al.* [22], Mallet-Paret & Sell [23] and Chow *et al.* [24] studied the inertial manifolds for nonlinear evolution equations. One can also find the applications of invariant manifolds in linearization [25,26], singular perturbations [27] and chaos [28].

On the other hand, the differential equations with non-instantaneous impulsive effect can be used to describe the periodic evolution of human in hemodynamic equilibrium. This new type of impulsive equation was first introduced in Hernández & O'Regan [29], which is a generalization of the impulsive differential equations (IDEs). However, it can be quite different from the instantaneous IDEs. It is well known that the classical IDE has an instantaneous jump at the impulsive points. But the difference is that the non-instantaneous impulsive effect starts at an arbitrary impulsive point and remains active on a finite time interval. We give a simple example to illustrate the difference between the non-instantaneous IDEs and the instantaneous IDEs. Consider the following non-instantaneous impulsive system

$$\begin{cases} \dot{x}(t) = x(t), & t \in (0,1] \cup (2,3] \cup (4,5] \cup (6,7] \cdots, \\ x(t_i^+) = 2x(t_i^-), & t_i = 1,3,5,7 \cdots, \\ x(t) = \varepsilon x(t_1^-), & t \in (1,2] \cup (3,4] \cup (5,6] \cup (7,8] \cdots, \\ x(s_j^+) = x(s_j^-), & s_j = 2,4,6,8 \cdots, \end{cases}$$

with the initial value $x(0) = 1$, where $t_i$ is impulsive point, $s_j$ is junction point and $0 < \varepsilon < 1$ is sufficiently small. Then the solution of the non-instantaneous impulsive system is given by

$$x(t) = e + 2\varepsilon e + e \cdot 2\varepsilon e + \cdots = \sum_{n=1}^{\infty} (1 + 2^n \varepsilon^n e^{n-1} + 2^n \varepsilon^n e^n) e.$$

If we take $\varepsilon < (1/4e)$, then $x(t)$ is bounded. For comparison, now we consider the instantaneous (classical) impulsive system with the same initial value $x(0) = 1$:

$$\begin{cases} \dot{x}(t) = x(t), & t \in (0,\infty), t \neq 1,2,3,\ldots, \\ x(t_i^+) = 2x(t_i^-), & t_i = 1,2,3,\ldots. \end{cases}$$

Then the solution of the instantaneous impulsive system is given by

$$x(t) = e + 2 e^2 + 4 e^3 + \cdots = \sum_{n=1}^{\infty} 2^{n-1} e^n.$$

Clearly, the solution of the instantaneous impulsive system $x(t)$ is unbounded. It is obvious that the buffering phase of the impulses have a great effect on the solution. The boundedness and stability of the solution can switch if the instantaneous impulsive effect is changed to the non-instantaneous impulsive effect. Therefore, it is meaningful to study stability theory and qualitative properties of non-instantaneous IDEs. Also in 2013, based on analytic semigroup and fixed point theory, Pierri *et al.* [30] obtained a novel result of the existence of the mild solution in fractional power space. Later, Hernández *et al.* [31] derived the existence of the mild and classical solution, and presented some applications involving partial non-instantaneous

IDEs. Feĉkan *et al.* [32] studied the existence of periodic solution for this new type of nonlinear evolution IDEs. Pierri *et al.* [33] proved the existence of the mild solutions and the asymptotical periodic solutions for a class of non-instantaneous IDEs. Recently, Wang *et al.* [34] considered the fractional non-instantaneous IDEs and studied the Cauchy problem of the fractional IDEs. Abbas & Benchohra [35] studied the Ulam stability of the partial fractional non-instantaneous IDEs. Gautam & Dabas [36] derived the mild solution of neutral fractional functional non-instantaneous IDEs. Furthermore, Colao *et al.* [37] pay attention to the delay effect on the non-instantaneous IDEs. Bai *et al.* [38] established a non-instantaneous pulse vaccination model to characterize the attractiveness of the infection-free periodic solution and the permanence of some sub-population. Hernández [39] studied a general class of non-instantaneous abstract impulsive problem 'without predefined times of impulse'. More recently, stability and robustness for non-instantaneous IDEs were given in Wang *et al.* [40–42] and Yang *et al.* [43]. The existence of an inertial manifold for semilinear non-instantaneous parabolic IDEs was given in Yang *et al.* [44]. The concept of a weak solution for non-instantaneous IDEs was introduced in Bai & Nieto [45]. Based on the classical Lax–Milgram Theorem, Bai and Nieto discussed the variational structure of the problem and the existence and uniqueness of weak solutions. Furthermore, another concept on non-instantaneous IDEs has been reported in the monograph of Agarwal *et al.* [46].

## (b) Basic notations and concepts

Let $\mathcal{B}(X)$ be the set of the bounded linear operator in Banach space $X$. Consider the following non-instantaneous linear impulsive evolution differential equations:

$$
\begin{cases}
\dot{y}(t) = A(t)y(t), & t \in [s_{i-1}, t_i], \quad i \in \mathbb{N}^+, \\
y(t_i^+) = B_i(t_i)y(t_i^-), & i \in \mathbb{N}^+, \\
y(t) = B_i(t)y(t_i^-), & t \in (t_i, s_i], \quad i \in \mathbb{N}^+, \\
y(s_i^+) = y(s_i^-), & i \in \mathbb{N}^+,
\end{cases}
\tag{1.1}
$$

for $\mathbb{N}^+ = \{1, 2, \ldots\}$. The impulsive points $t_i$ and junction points $s_i$ satisfy $s_0 = 0$ and $s_{i-1} < t_i < s_i < t_{i+1} < \cdots$, for all $i = \{1, 2, \ldots\}$, in addition $t_i \to \infty$ as $i \to \infty$. The symbols $y(\varsigma_i^+)$ and $y(\varsigma_i^-)$ represent the right and left limits of $y(t)$ at $t = \varsigma_i$, define $y(\varsigma_i^-) = y(\varsigma_i)$. Let $\mathbb{T} = \bigcup_{i=1}^{\infty}[s_{i-1}, t_i]$ and $\mathbb{J} = \bigcup_{i=1}^{\infty}(t_i, s_i]$. Thus $\mathbb{R}_+ = \mathbb{T} + \mathbb{J}$. Assume that $A(\cdot) : \mathbb{T} \to \mathcal{B}(X)$, $B_i(\cdot) : \mathbb{J} \to \mathcal{B}(X)$. Moreover,

$$
0 < |s_i - t_i| = \theta_i \le \theta, \ i \in \mathbb{N}^+ \quad \text{and} \quad \limsup_{t \to \infty} \frac{\operatorname{card}\{i \in \mathbb{N}^+ : |t_i| < t\}}{t} = \rho < \infty.
$$

These conditions ensure the existence and uniqueness of the global right-continuous solution of (1.1). Let $W(t, s)$ be the evolution operator satisfying $y(t) = W(t, s)y(s)$ for each solution $y(t)$ of (1.1) for all $t, s \in \mathbb{R}_+$. Clearly, $W(t, s)W(s, \tau) = W(t, \tau)$ and $W(t, t) = \operatorname{id}_X$ for all $t \ge s \ge \tau$ with $t, s, \tau \in \mathbb{R}_+$, where $\operatorname{id}_X$ denotes identity operator.

Next, we present a definition of exponential dichotomy.

**Definition 1.1.** We say that (1.1) admits an exponential dichotomy on $\mathbb{R}_+$, if there exists projection $P(t)$ for each $t \in \mathbb{R}_+$ satisfying

$$
W(t, s)P(s) = P(t)W(t, s), \quad t \ge s \ge 0,
$$

and there exist constants $k, \alpha > 0$ such that

$$
\begin{cases}
\|W(t, s)P(s)\| \le k\,\mathrm{e}^{-\alpha(t-s)}, & t \ge s \ge 0, \\
\|W(s, t)^{-1}Q(s)\| \le k\,\mathrm{e}^{-\alpha(s-t)}, & s \ge t \ge 0,
\end{cases}
$$

where $Q(t) = \operatorname{id}_X - P(t)$ is the complementary projection of $P(t)$.

In the present paper, we establish the theory of stable manifold based on the classical method of Lyapunov–Perron and Hadamard. Assume that the non-instantaneous linear impulsive differential equation admits an exponential dichotomy, we give the conditions of the existence of

the global and local stable manifolds. Furthermore, $C^k$-smoothness and periodicity of the stable manifold is obtained. We consider the following semilinear non-instantaneous IDEs:

$$\begin{cases} y'(t) = A(t)y(t) + f(t, y(t)), & t \in \mathbb{T}, \\ y(t_i^+) = B_i(t_i)y(t_i^-) + g_i(y(t_i^-)), & i \in \mathbb{N}^+, \\ y(t) = B_i(t)y(t_i^-) + g_i(y(t_i^-)), & t \in \mathbb{J}, \\ y(s_i^+) = y(s_i^-), & i \in \mathbb{N}^+, \end{cases} \tag{1.2}$$

where the nonlinearities $f : \mathbb{T} \times X \to X$, $g : X \to X$ are both piecewise continuous. Denote $r(t, \tau)$ by the number of impulsive points existing in the interval $(\tau, t)$. Now, the mild solution $y(\cdot)$ to (1.2) satisfies the following integral equation:

$$y(t) = W(t, \tau)y(\tau) + \int_\tau^t W(t, s)\bar{f}(s, y(s)) \, ds + \sum_{i=1}^{r(t,\tau)} W(t, s_i)g_i(y(t_i^-)), \quad \text{for } t \geq \tau, \tag{1.3}$$

where $\tau \in [0, t_1]$, $W(t, s_i) = W(t, t_i^+)$ and

$$\bar{f}(t, y) = \begin{cases} f(t, y), & t \in \mathbb{T}, \\ 0, & t \in \mathbb{J}. \end{cases}$$

## (c) The strength of the non-instantaneous IDEs

We believe that the non-instantaneous IDEs are very complex and generalized, which includes the following special cases:

1. If the algebraic equation of equation (1.1) disappears, then equation (1.1) reduces to the instantaneous impulsive equation. Furthermore, there are two cases:
   — Assume that the differential equation also disappears, the instantaneous impulsive equation reduces to the difference equation;
   — Assume that the impulsive point $t_i$ disappears, then the instantaneous impulsive equation reduces to a general continuous ODE.
2. If the differential equation of equation (1.1) disappears, then equation (1.1) reduces to a piecewise algebraic equation. Similarly, there are also two cases: (1) difference equation and (2) continuous algebraic equation.

## (d) Outline of this paper

The rest of the present paper is organized as follows. In §2, we present the existence of global and local stable manifolds. In §3, the $C^k$-smoothness of the stable manifold is given. In §4, the periodicity of the stable manifold is given. Finally, as an application, we obtain the existence of stable manifold for the non-instantaneous impulsive Duffing oscillators.

## 2. Existence of stable manifolds

Let $X$ be a Banach space and let $\mathbb{I} \subset \mathbb{R}_+$ be an interval. Let

$PC(\mathbb{I}, X) := \{x : \mathbb{I} \to X | x \in C((t_i, t_{i+1}], X), \ x(t_i^+), x(t_i^-) \text{ exist for each } i \in \mathbb{N}^+ \text{ and } \sup ||x(t)|| < \infty\}$

and $||x|| := \sup\{||x(t)|| < \infty | t \in \mathbb{I}\}$. Let

$PC_\rho(\mathbb{I}, X) := \{x : \mathbb{I} \to X | x \in C((t_i, t_{i+1}], X), \ x(t_i^+), x(t_i^-) \text{ exist for each } i \in \mathbb{N}^+ \text{ and } \sup ||x(t)|| \, e^{-\rho t} < \infty\}$

and $||x||_{PCB} := \sup\{||x(t)|| \, e^{-\rho t} < \infty | t \in \mathbb{I}, \rho \in \mathbb{R}\}$.

Obviously, $(PC(\mathbb{I}, X), ||x||)$ and $(PC_\rho(\mathbb{I}, X), ||x||_{PCB})$ are both Banach spaces with norms $|| \cdot ||$ and $|| \cdot ||_{PCB}$, respectively. Clearly, $(PC_\rho(\mathbb{I}, X), ||x||_{PCB})$ is a more general Banach space than the space $(PC(\mathbb{I}, X), ||x||)$.

We consider the functions $\varXi_j(\cdot): \mathbb{T} \to \mathbb{R}, j = 1, 2, 3$ defined by

$$\varXi_1(t) := \sum_{i=1}^{r(t,\tau)} e^{-\alpha(s_i-\tau)+\rho t_i^-}, \quad \varXi_2(t) := \sum_{i=1}^{r(t,\tau)} e^{-(\alpha+\rho)(t-s_i)}$$

and

$$\varXi_3(t) := \sum_{i=r(t,\tau)+1}^{\infty} e^{(\alpha+\rho)(t-s_i)}.$$

If $r(t, \tau) = 0$ (i.e. there is no impulsive point on $(\tau, t)$), then $\varXi_j(t) \equiv 0$. In addition, taking $R_j := \sup_{t \geq \tau} \varXi_j(t) < \infty$.

Throughout this paper, we make the following assumptions.

$(H_1)$ linear equation (1.1) admits an exponential dichotomy on $\mathbb{R}_+$ with $k, \alpha > 0$ and projections $P(t), Q(t)$;

$(H_2)$ there exist $\mathrm{Lip}(\bar{f}) > 0$ and $\mathrm{Lip}(g_i) > 0$ such that

$$||\bar{f}(t, u) - \bar{f}(t, v)|| \leq \mathrm{Lip}(\bar{f})||u - v|| \quad \text{and} \quad ||g_i(u) - g_i(v)|| \leq \mathrm{Lip}(g_i)||u - v||,$$

for all $t \in \mathbb{R}_+$ and $u, v \in X$. Moreover, $\bar{f}(t, 0) \equiv 0, g_i(0) \equiv 0$;

$(H_2')$ there exists a continuous and non-decreasing mapping $L(s): [0, +\infty) \to [0, +\infty)$ such that

$$||\bar{f}(t, u) - \bar{f}(t, v)|| \leq L(\max\{||u||, ||v||\})||u - v||$$

and

$$||g_i(u) - g_i(v)|| \leq L(\max\{||u||, ||v||\})||u - v||,$$

and $L(0) \equiv 0, \bar{f}(t, 0) \equiv 0, g_i(0) \equiv 0$.

Now, we present our theorems on the existence of stable manifolds of system (1.2).

**Theorem 2.1 (Global stable manifold).** *Consider system (1.2) with assumptions $(H_1)$ and $(H_2)$ in Banach space $PC(\mathbb{I}, X)$. If $|\rho| < \alpha$, and*

$$\mathcal{L} := \left\{ \frac{k \cdot \mathrm{Lip}(\bar{f})}{\alpha - \rho} + \frac{k \cdot \mathrm{Lip}(\bar{f})}{\alpha + \rho} + k \cdot \mathrm{Lip}(g_i)(R_2 + R_3) \right\} < 1, \tag{2.1}$$

*then the following results hold:*

(i) *system (1.2) has a global stable manifold:*

$$\mathcal{M} := \{y(\tau) | y(t; \tau, y(\tau)) \text{ is defined in } PC_\rho(\mathbb{I}, X)\};$$

(ii) *$\mathcal{M} := \{\xi + h(\tau, \xi) | \xi \in \mathcal{R}P(\tau)\}$, where $h(\tau, \cdot): \mathcal{R}P(\tau) \to \mathcal{R}Q(\tau)$ is Lipschitz continuous in the norm $|| \cdot ||$ and $h(\tau, 0) \equiv 0$.*

**Theorem 2.2 (Local stable manifold).** *Consider system (1.2) with assumptions $(H_1)$ and $(H_2')$ in Hilbert space $X$. Let $|| \cdot ||$ be the norm induced by the inner product $\langle \cdot, \cdot \rangle$. Then system (1.2) has a local stable manifold*

$$\mathcal{M}_{\mathrm{loc}} := \{\xi + h(\tau, \xi) | \xi \in \mathcal{R}P(\tau) \cap B(0, \bar{r})\},$$

*where $B(0, \bar{r})$ is a spherical neighbourhood and $\bar{r} = \min\{1, 1/k\}r/2$. Furthermore,*

(i) *$h(\tau, \xi)$ is strongly continuous in $\tau$ and Lipschitz continuous in $\xi$ in the norm $|| \cdot ||$, and $\mathrm{Lip}(h) \leq \frac{1}{2}, h(\tau, 0) \equiv 0$;*

(ii) *$\mathcal{M}_{\mathrm{loc}}$ is homeomorphic with an open ball in $\mathcal{R}P(\tau)$;*

(iii) *$\mathcal{M}_{\mathrm{loc}}$ is tangent to $\mathcal{R}P(\tau)$ at $\xi = 0$;*

*where $r$ is called the locality radius of the stable manifold $\mathcal{M}$ if $r$ is the maximum radius of local coordinate charts of the manifold $\mathcal{M}$ at $0 \in \mathcal{M}$.*

If the non-instantaneous impulsive effects reduce to the instantaneous impulsive effects, then the following corollary follows.

**Corollary 2.3.** *Suppose that system (1.2) reduces to the instantaneous impulsive equation, namely, impulsive effect start at an arbitrary point but it does not remain active on finite time intervals again. If assumptions ($H_1$), ($H_2$) and (2.1) hold, then the Lipschitzian stable manifold is given by*

$$\mathcal{M}_1 := \{\xi + h_1(\tau, \xi) | \xi \in \mathcal{R}P(\tau)\},$$

*where*

$$h_1(\tau, \xi) = -\int_\tau^\infty W(\tau, s)Q(s)\bar{f}(s, \varphi_\tau(\xi)(s))\, ds - \sum_{\tau \leq s_i} W(\tau, s_i)Q(s_i)g_i(\varphi_\tau(\xi)(s_i)).$$

Note that here $\bar{f}$ is piecewise continuous on $\mathbb{R}_+ \backslash \{t_i\}_{i \in \mathbb{N}}$ and $g_i$ is defined in $\{t_i\}_{i \in \mathbb{N}}$.

If the impulsive effects are absent, then the following corollary follows.

**Corollary 2.4.** *If the impulsive effects of system (1.2) are absent, further we assume that assumption ($H_1$) holds and the nonlinear term f satisfies Lipschitz continuous and*

$$\mathcal{L} := \frac{k \cdot Lip(\bar{f})}{\alpha - \rho} + \frac{k \cdot Lip(\bar{f})}{\alpha + \rho} < 1. \tag{2.2}$$

*Then the Lipschitzian stable manifold is given by*

$$\mathcal{M}_2 := \{\xi + h_2(\tau, \xi) | \xi \in \mathcal{R}P(\tau)\},$$

*where*

$$h_2(\tau, \xi) = -\int_\tau^\infty W(\tau, s)Q(s)f(s, \varphi_\tau(\xi)(s))\, ds.$$

**Remark 2.5.** When the impulsive effects are absent, we generalize and improve some known results on the stable manifolds in the literature. For example, if $\rho = 0$, then corollary 2.4 reduces to the classical invariant manifold theory (see ch. 2, Perko [47], pp. 104–118). Furthermore, the condition (i.e. (2.2)) of corollary 2.4 is consistent with Zhang [13]. Based on a more general Banach space $(PC_\rho(\mathbb{I}, X), ||x||_{PCB})$, our results are new and different from [8,10,11].

To prove our theorem, we need the following lemmas.

**Lemma 2.6.** *Suppose that ($H_1$), ($H_2$) hold. If (1.3) has a solution $y(t; \tau, y(\tau)) \in PC_\rho(\mathbb{I}, X)$, then we have the following expression:*

$$y(t) = W(t, \tau)P(\tau)y(\tau) + \int_\tau^t W(t, s)P(s)\bar{f}(s, y(s))\, ds + \sum_{i=1}^{r(t,\tau)} W(t, s_i)P(s_i)g_i(y(t_i^-))$$

$$- \int_t^\infty W(t, s)Q(s)\bar{f}(s, y(s))\, ds - \sum_{i=r(t,\tau)+1}^\infty W(t, s_i)Q(s_i)g_i(y(t_i^-)). \tag{2.3}$$

*Proof.* We split $y(t; \tau, y(\tau))$ into two parts: $P(t)y(t; \tau, y(\tau))$ and $Q(t)y(t; \tau, y(\tau))$. Then

$$P(t)y(t) = P(t)W(t, \tau)y(\tau) + \int_\tau^t P(t)W(t, s)\bar{f}(s, y(s))\, ds + \sum_{i=1}^{r(t,\tau)} P(t)W(t, s_i)g_i(y(t_i^-))$$

$$= W(t, \tau)P(\tau)y(\tau) + \int_\tau^t W(t, s)P(s)\bar{f}(s, y(s))\, ds + \sum_{i=1}^{r(t,\tau)} W(t, s_i)P(s_i)g_i(y(t_i^-)) \tag{2.4}$$

and

$$Q(t)y(t) = W(t, \tau)Q(\tau)y(\tau) + \int_\tau^t W(t, s)Q(s)\bar{f}(s, y(s))\, ds + \sum_{i=1}^{r(t,\tau)} W(t, s_i)Q(s_i)g_i(y(t_i^-)). \tag{2.5}$$

It is not difficult to obtain that

$$Q(\tau)y(\tau) = W(\tau,t)Q(t)y(t) - \int_\tau^t W(\tau,s)Q(s)\bar{f}(s,y(s))\,\mathrm{d}s - \sum_{i=1}^{r(t,\tau)} W(\tau,s_i)Q(s_i)g_i(y(t_i^-)).$$

Since $y(t) \in PC_\rho(\mathbb{I}, X)$, we have $\sup_{t\in\mathbb{I}}\{e^{-\rho t}||y(t)||\} < \infty$, denoted by $||y||_{PCB}$. Thus

$$||W(\tau,t)Q(t)y(t)|| \le k\,e^{-\alpha(t-\tau)} \cdot ||y(t)|| \le k\,e^{(\rho-\alpha)t+\alpha\tau} \cdot ||y||_{PCB},$$

$$\left\|\int_\tau^t W(\tau,s)Q(s)\bar{f}(s,y(s))\,\mathrm{d}s\right\| \le \int_\tau^t k\,e^{-\alpha(s-\tau)} \cdot \mathrm{Lip}(\bar{f}) \cdot ||y(s)||\,\mathrm{d}s$$

$$\le k \cdot \mathrm{Lip}(\bar{f}) \cdot ||y||_{PCB} \cdot \frac{e^{\rho\tau}}{\alpha - \rho},$$

and

$$\left\|\sum_{i=1}^{r(t,\tau)} W(\tau,s_i)Q(s_i)g_i(y(t_i^-))\right\| \le \sum_{i=1}^{r(t,\tau)} k\,e^{-\alpha(s_i-\tau)} \cdot \mathrm{Lip}(g_i) \cdot ||y(t_i^-)||$$

$$\le k \cdot \mathrm{Lip}(g_i) \cdot ||y||_{PCB} \cdot \sum_{i=1}^{r(t,\tau)} e^{-\alpha(s_i-\tau)+\rho t_i^-}$$

$$\le k \cdot \mathrm{Lip}(g_i) \cdot ||y||_{PCB} \cdot R_1.$$

Note that $|\rho| < \alpha$ and $\tau \in [0, t_1]$, if we take $t \to \infty$, we have

$$Q(\tau)y(\tau) = -\int_\tau^\infty W(\tau,s)Q(s) \cdot \bar{f}(s,y(s))\,\mathrm{d}s - \sum_{i=1}^\infty W(\tau,s_i)Q(s_i)g_i(y(t_i^-)).$$

Hence, it follows from (2.5) that

$$Q(t)y(t) = -\int_t^\infty W(t,s)Q(s)\bar{f}(s,y(s))\,\mathrm{d}s - \sum_{i=r(t,\tau)+1}^\infty W(t,s_i)Q(s_i)g_i(y(t_i^-)).$$

Combining this equality with (2.4), lemma 2.6 follows. ∎

**Lemma 2.7.** *Suppose that* $(H_1)$ *and* $(H_2)$ *hold. If (2.1) holds for any* $\tau \in [0, t_1]$ *and* $\xi \in \mathscr{R}P(\tau)$, *then integral equation (1.3) with the initial value* $P(\tau)y(\tau) = \xi$ *has a unique solution* $\varphi_\tau(\xi)$ *in* $PC_\rho(\mathbb{I}, X)$. *Moreover,* $\varphi_\tau(\cdot)$ *is Lipschitz continuous in the norm* $||\cdot||$ *and if* $\xi \equiv 0$, $\varphi_\tau(0) \equiv 0$.

*Proof.* For convenience, we write

$$(\mathscr{J}_1\bar{f})(t) = \int_\tau^t W(t,s)P(s)\bar{f}(s)\,\mathrm{d}s - \int_t^\infty W(t,s)Q(s)\bar{f}(s)\,\mathrm{d}s$$

and

$$(\mathscr{J}_2 g_k)(t_i) = \sum_{i=1}^{r(t,\tau)} W(t,s_i)P(s_i)g_i(t_i) - \sum_{i=r(t,\tau)+1}^\infty W(t,s_i)Q(s_i)g_i(t_i).$$

Obviously, $\mathscr{J}_1$ and $\mathscr{J}_2$ are linear operator. Set $\mathscr{F}(y)(s) = \bar{f}(s,y(s))$ and $\mathscr{G}(y)(t_i^-) = g_i(y(t_i^-))$. Then equation (2.3) can be rewritten as follows:

$$y(t) = W(t,\tau)P(\tau)P(\tau)y(\tau) + \mathscr{J}_1(\mathscr{F}(y))(t) + \mathscr{J}_2(\mathscr{G}(y))(t_i^-). \tag{2.6}$$

For any $y(t) \in PC_\rho(\mathbb{I}, X)$, we define

$$(\mathscr{T}_\xi y)(t) := W(t,\tau)P(\tau)\xi + \mathscr{J}_1(\mathscr{F}(y))(t) + \mathscr{J}_2(\mathscr{G}(y))(t_i^-). \tag{2.7}$$

Now we claim that $\mathscr{T}_\xi$ is a contraction mapping from $PC_\rho(\mathbb{I}, X)$ into $PC_\rho(\mathbb{I}, X)$. Firstly, we say that $\mathscr{T}_\xi$ maps $PC_\rho(\mathbb{I}, X)$ into itself. In fact,

$$
\begin{aligned}
&||\mathscr{J}_1(\mathscr{F}(y))(t)||\,e^{-\rho t} \\
&\le e^{-\rho t}\left\{\int_\tau^t ||W(t,s)P(s)\bar{f}(s,y(s))||\,ds + \int_t^\infty ||W(t,s)Q(s)\bar{f}(s,y(s))||\,ds\right\} \\
&\le e^{-\rho t}\left\{\int_\tau^t k\,e^{-\alpha(t-s)}\cdot\mathrm{Lip}(f)\cdot||y(s)||\,ds + \int_t^\infty e^{\alpha(t-s)}\cdot\mathrm{Lip}(\bar{f})\cdot||y(s)||\,ds\right\} \\
&\le k\cdot\mathrm{Lip}(\bar{f})\cdot||y||_{PCB}\cdot\left\{\int_\tau^t e^{-(\alpha+\rho)(t-s)}ds + \int_t^\infty e^{(\alpha-\rho)(t-s)}ds\right\} \\
&\le k\cdot\mathrm{Lip}(\bar{f})\cdot\left\{\frac{1}{\alpha+\rho} + \frac{1}{\alpha-\rho}\right\}||y||_{PCB}
\end{aligned}
$$

and

$$
\begin{aligned}
&||\mathscr{J}_2(\mathscr{G}(y))(t_i^-)||\,e^{-\rho t} \\
&\le e^{-\rho t}\left\{\sum_{i=1}^{r(t,\tau)} ||W(t,s_i)P(s_i)g_i(y(t_i^-))|| + \sum_{i=r(t,\tau)+1}^\infty ||W(t,s_i)Q(s_i)g_i(y(t_i^-))||\right\} \\
&\le e^{-\rho t}\left\{\sum_{i=1}^{r(t,\tau)} k\,e^{-\alpha(t-s_i)}\cdot\mathrm{Lip}(g_i)\cdot||y(t_i^-)|| + \sum_{i=r(t,\tau)+1}^\infty k\,e^{\alpha(t-s_i)}\cdot\mathrm{Lip}(g_i)\cdot||y(t_i^-)||\right\} \\
&\le k\cdot\mathrm{Lip}(g_i)\cdot||y||_{PCB}\left\{\sum_{i=1}^{r(t,\tau)} e^{-(\alpha+\rho)(t-s_i)} + \sum_{i=r(t,\tau)+1}^\infty e^{(\alpha-\rho)(t-s_i)}\right\} \\
&\le k\cdot\mathrm{Lip}(g_i)\cdot||y||_{PCB}\cdot(R_2+R_3).
\end{aligned}
$$

Thus,

$$
\begin{aligned}
&||(\mathscr{T}_\xi y)(t)||\,e^{-\rho t} \\
&\le k\left\{e^{-(\alpha+\rho)t+\alpha\tau}||\xi|| + \left[\frac{\mathrm{Lip}(\bar{f})}{\alpha-\rho} + \frac{\mathrm{Lip}(\bar{f})}{\alpha+\rho} + \mathrm{Lip}(g_i)(R_2+R_3)\right]||y||_{PCB}\right\} < \infty.
\end{aligned}
$$

Next, we are going to prove that $\mathscr{T}_\xi$ is a contraction uniformly with respect to $\xi$. Note that $\mathcal{L} = \{(k\cdot\mathrm{Lip}(\bar{f})/(\alpha-\rho)) + (k\cdot\mathrm{Lip}(\bar{f})/(\alpha+\rho)) + k\cdot\mathrm{Lip}(g_i)(R_2+R_3)\} < 1$, for any $y_1, y_2 \in PC_\rho(\mathbb{I}, X)$ and $\xi \in \mathscr{R}P(\tau)$, we have

$$
\begin{aligned}
||\mathscr{T}_\xi y_1 - \mathscr{T}_\xi y_2||_{PCB} &\le ||\mathscr{J}_1(\mathscr{F}(y_1)) - \mathscr{J}_1(\mathscr{F}(y_2))||_{PCB} + ||\mathscr{J}_2(\mathscr{G}(y_1)) - \mathscr{J}_2(\mathscr{G}(y_2))||_{PCB} \\
&\le \mathcal{L}\cdot||y_1 - y_2||_{PCB}.
\end{aligned}
$$

Hence, from (2.1), $\mathscr{T}_\xi$ is a contraction mapping in $PC_\rho(\mathbb{I}, X)$, namely, it has a unique fixed point $\varphi_\tau(\xi)(t) \in PC_\rho(\mathbb{I}, X)$. Moreover, for any $\xi_1, \xi_2 \in \mathscr{R}P(\tau)$ and $y(t) \in PC_\rho(\mathbb{I}, X)$,

$$
\begin{aligned}
||\mathscr{T}_{\xi_1}y - \mathscr{T}_{\xi_2}y||_{PCB} &\le ||W(t,\tau)P(\tau)(\xi_1 - \xi_2)||\,e^{-\rho t} \\
&\le k\,e^{-(\alpha+\rho)(t-\tau)}\cdot e^{-\rho\tau}||\xi_1 - \xi_2|| \\
&\le k\,e^{-\rho\tau}||\xi_1 - \xi_2||.
\end{aligned}
$$

It is clear that

$$||\varphi_\tau(\xi_1) - \varphi_\tau(\xi_2)||_{PCB}$$

$$= ||\mathscr{T}_{\xi_1}(\varphi_\tau(\xi_1)) - \mathscr{T}_{\xi_2}(\varphi_\tau(\xi_2))||_{PCB}$$

$$\leq ||\mathscr{T}_{\xi_1}(\varphi_\tau(\xi_1)) - \mathscr{T}_{\xi_1}(\varphi_\tau(\xi_2))||_{PCB} + ||\mathscr{T}_{\xi_1}(\varphi_\tau(\xi_2)) - \mathscr{T}_{\xi_2}(\varphi_\tau(\xi_2))||_{PCB}$$

$$\leq \mathcal{L} \cdot ||\varphi_\tau(\xi_1) - \varphi_\tau(\xi_2)||_{PCB} + k\,e^{-\rho\tau}||\xi_1 - \xi_2||.$$

It follows from (2.1) that

$$||\varphi_\tau(\xi_1) - \varphi_\tau(\xi_2)||_{PCB} \leq \frac{k\,e^{-\rho\tau}}{1 - \mathcal{L}} \cdot ||\xi_1 - \xi_2||. \tag{2.8}$$

Therefore, $\varphi_\tau(\cdot)$ is Lipschitz continuous in the norm $||\cdot||$. If $\xi = 0$, by the uniqueness of solution of (1.2) we see $\varphi_\tau(0) = 0$. We complete the proof of lemma 2.7. ∎

**Lemma 2.8.** *For any $|\rho| < \alpha$ and $\tau \in [0, t_1]$, $y(t) \in PC_\rho(\mathbb{I}, X)$ iff $y(\cdot + \tau) \in PC_\rho([0, +\infty), X)$.*

*Proof.* Denoted $||\cdot||_{PCB}$ and $||\cdot||_{PCB^0}$ by the norm of $PC_\rho(\mathbb{I}, X)$ and $PC_\rho([0, +\infty), X)$, respectively. It is easy to see that

$$||y(\cdot + \tau)||_{PCB^0} = \sup_{t \in [0, +\infty)} e^{-\rho t}||y(t + \tau)|| = e^{\rho\tau} \sup_{t \in [0, +\infty)} e^{-\rho(t+\tau)}||y(t + \tau)||$$

$$= e^{\rho\tau} \sup_{s \in \mathbb{I}} e^{-\rho s}||y(s)|| = e^{\rho\tau}||y(\cdot)||_{PCB}.$$

∎

Let $\sigma$ be a $C^\infty$ cut-off function from $[0, \infty)$ to $[0, 1]$, i.e. if $0 \leq s \leq 1$, $\sigma(s) = 1$; if $s \geq 2$, $\sigma(s) = 0$; if $1 < s < 2$, $0 < \sigma(s) < 1$ and $|\sigma'(s)| \leq 2$. In order to prove theorem 2.2, firstly, we are going to discuss the modified equation of (1.2), that is,

$$\begin{cases} \dot{y}(t) = A(t)y(t) + \bar{f}_r(t, y(t)), & t \in \mathbb{T}, \\ y(t_i^+) = B_i(t_i)y(t_i^-) + g_{i,r}(y(t_i^-)), & i \in \mathbb{N}^+, \\ y(t) = B_i(t)y(t_i^-) + g_{i,r}(y(t_i^-)), & t \in \mathbb{J}, \\ y(s_i^+) = y(s_i^-), & i \in \mathbb{N}^+, \end{cases} \tag{1.2r}$$

where $r > 0$ is a given constant. $\bar{f}_r(t, y(t)), g_{i,r}(y(t_i^-))$ are the modified mappings of $\bar{f}(t, y(t))$ and $g_i(y(t_i^-))$, respectively, defined as follows:

$$\bar{f}_r(t, y(t)) = \bar{f}\left(t, \sigma\left(\frac{||y||}{r}\right)y(t)\right) \quad \text{and} \quad g_{i,r}(y(t_i^-)) = g_i\left(\sigma\left(\frac{||y||}{r}\right)y(t_i^-)\right).$$

Obviously, $\sigma(||y||)$ is a smooth cut-off function in Hilbert space $X$, and

$$\left|\left|D_y\left(\sigma\left(\frac{||y||}{r}\right)y\right)\right|\right| \leq \sigma\left(\frac{||y||}{r}\right) + ||y|| \cdot \left|\left|\sigma'\left(\frac{||y||}{r}\right)\right|\right| \cdot \frac{1}{r} \leq 1 + 2r \cdot 2 \cdot \frac{1}{r} = 5.$$

Then the modified mappings $\bar{f}_r(t, y(t)), g_{i,r}(y(t_i^-))$ have the following properties:

(1) $\bar{f}_r(t, y(t))|\bar{B}(0, r) \equiv \bar{f}(t, y)$, $\quad \bar{f}_r(t, y(t))|\{y \in X | ||y|| > 2r\} \equiv 0$, $\quad g_{i,r}(y(t_i^-))|\bar{B}(0, r) \equiv g_i(y(t_i^-))$, $g_{i,r}(y(t_i^-))|\{y \in X | ||y|| > 2r\} \equiv 0$, where $\bar{B}(0, r)$ is the closure of $B(0, r)$;

(2) $||\bar{f}_r(t, u) - \bar{f}_r(t, v)|| \leq 5 \cdot L(2r)||u - v||$, $||g_{i,r}(u(t_i)) - g_{i,r}(v(t_i))|| \leq 5 \cdot L(2r)||u - v||$, where $t \in \mathbb{R}_+, i \in \mathbb{N}^+, u, v \in X$.

Now we are in a position to prove theorems 2.1 and 2.2.

*Proof of theorem 2.1.* For any $\tau \in [0, t_1]$ and $\xi \in \mathscr{R}P(\tau)$, it follows from lemma 2.7 that there exists a unique solution $\varphi_\tau(\xi)$ in $PC_\rho(\mathbb{I}, X)$ such that $\varphi_\tau(0) = 0$ and

$$\varphi_\tau(\xi)(t) = W(t, \tau)P(\tau)\xi + \mathscr{J}_1(\mathscr{F}(\varphi_\tau(\xi)))(t) + \mathscr{J}_2(\mathscr{G}(\varphi_\tau(\xi)))(t_i^-). \tag{2.9}$$

Let

$$\mathscr{M} = \{y(\tau) | y(t; \tau, y(\tau)) \text{ is defined in } PC_\rho(\mathbb{I}, X)\}.$$

From lemma 2.6 and (2.9), it is clear that the initial values $y(\tau)$, which compose the set $\mathscr{M}$ can be written as follows:

$$\begin{aligned}
y(\tau) &= \varphi_\tau(\xi)(\tau) = \xi + \mathscr{J}_1(\mathscr{F}(\varphi_\tau(\xi)))(\tau) + \mathscr{J}_2(\mathscr{G}(\varphi_\tau(\xi)))(t_i^-) \\
&= \xi - \int_\tau^\infty W(\tau, s)Q(s)\bar{f}(s, \varphi_\tau(\xi)(s)) \, ds - \sum_{i=1}^\infty W(\tau, s_i)Q(s_i)g_i(\varphi_\tau(\xi)(t_i^-)) \\
&\triangleq \xi + h(\tau, \xi),
\end{aligned}$$

where

$$h(\tau, \xi) = -\int_\tau^\infty W(\tau, s)Q(s)\bar{f}(s, \varphi_\tau(\xi)(s)) \, ds - \sum_{i=1}^\infty W(\tau, s_i)Q(s_i)g_i(\varphi_\tau(\xi)(t_i^-)).$$

Since $\varphi_\tau(\cdot)$ is Lipschitz in the norm $|| \cdot ||$, for any $\xi_1, \xi_2 \in \mathscr{R}P(\tau)$ we have

$$\begin{aligned}
||h(\tau, \xi_1) - h(\tau, \xi_2)||_{PCB} &\leq e^{-\rho\tau}\left\{ \int_\tau^\infty k\,e^{-\alpha(s-\tau)} \cdot \text{Lip}(\bar{f}) \cdot ||\varphi_\tau(\xi_1)(s) - \varphi_\tau(\xi_2)(s)|| \, ds \right. \\
&\quad \left. + \sum_{i=1}^\infty k\,e^{-\alpha(s_i-\tau)} \cdot \text{Lip}(g_i) \cdot ||\varphi_\tau(\xi_1)(t_i^-) - \varphi_\tau(\xi_2)(t_i^-)|| \right\} \\
&\leq \frac{k\,e^{-\rho\tau}\mathcal{L}}{1 - \mathcal{L}} \cdot ||\xi_1 - \xi_2||_{PCB}.
\end{aligned}$$

Hence $h(\tau, \xi)$ is Lipschitz in the norm $|| \cdot ||$ in $\xi$ for any $\tau \in [0, t_1]$. Moreover, $h(\tau, 0) = 0$, since $\varphi_\tau(0) = 0$ and $\bar{f}(t, 0) = 0, g_i(0) = 0$. Therefore, $\mathscr{M}$ is a stable manifold. To prove that $\mathscr{M}$ is invariant, take $(\tau, y(\tau)) \in \mathscr{M}$. Since $\bar{f}, g_i$ are Lipschitz on $\mathbb{R}_+$, the unique solution $y(t; \tau, y(\tau))$ is defined for all $t \in \mathbb{R}_+$ and $y(t; \tau, y(\tau)) \in PC_\rho(\mathbb{I}, X)$. Now we claim that $y(t; \tau, y(\tau)) \in PC_\rho(\mathbb{R}_+, X)$. In fact, for any given $0 \leq t^* < \tau$, let $y_1^* = y(t; \tau, y(\tau))$. Obviously, equation (1.2) has a unique solution $y(t; t^*, y_1^*)$ through $(t^*, y_1^*)$. It is clear to see that $(\tau, y(t; t^*, y_1^*)) = (\tau, y(\tau)) \in \mathscr{M}$ and $\sup_{t \in [t^*, \tau]} ||y(t; t^*, y_1^*)|| \leq M$ for some $M > 0$. Then, $y(t; t^*, y_1^*) \in PC_\rho([t^*, \infty), X)$. Since $t^* \in [0, \tau)$ is arbitrary, $y(t; \tau, y(\tau)) \in PC_\rho(\mathbb{R}_+, X)$. We complete the proof of theorem 2.1. ∎

*Proof of theorem 2.2.* Step 1: we prove (1) firstly. Choose a sufficiently small $r > 0$ such that

$$\left.\begin{aligned}
5k\left\{\frac{L(2r)}{\alpha - \rho} + \frac{L(2r)}{\alpha + \rho}\right\} &< \frac{1}{4}, \\
5k\{L(2r)(R_2 + R_3)\} &< \frac{1}{4} \\
10k^2\left\{\frac{L(2r)}{\alpha - \rho} + \frac{L(2r)}{\alpha + \rho}\right\} &< \frac{1}{2}.
\end{aligned}\right\} \tag{2.10}$$

and

Let $\bar{r} = \min\{1, 1/k\}r/2$. For $\xi \in \mathscr{R}(\tau) \cap B(0, \bar{r})$ and $\tau \in [0, t_1]$, we define

$$\mathscr{K}_\tau(\xi, r) := \{y \in PC_\rho(\mathbb{I}, X) | ||y(\cdot + \tau)||_{PCB^0} \leq r, P(\tau)y(\tau) = \xi\}.$$

It follows from lemma 2.8 that $\mathscr{K}_\tau(\xi, r)$ is a closed bounded subset in $PC_\rho(\mathbb{I}, X)$ in the topology induced by the norm $|| \cdot ||_{PCB^0}$. From the properties of $\bar{f}_r, g_{i,r}$, we know that $\text{Lip}(\bar{f}_r) \leq 5L(2r), \text{Lip}(g_{i,r}) \leq 5L(2r)$, and equation (1.2r) in $\bar{B}(0, r)$.

We consider the corresponding modified integral equation as follows:

$$y(t) = W(t,\tau)y(\tau) + \int_\tau^t W(t,s)\bar{f}_r(s,y(s))\,\mathrm{d}s + \sum_{i=1}^{r(t,\tau)} W(t,s_i)g_{i,r}(y(t_i^-)), \quad \text{for } t \geq \tau, \tag{2.1r}$$

We shall prove that equation (2.1r) has a unique solution in $\mathscr{K}_\tau(\xi,r)$. For any $y \in \mathscr{K}_\tau(\xi,r)$, define

$$(\mathscr{T}_\xi y)(t) := W(t,\tau)P(\tau)\xi + \mathscr{J}_1(\bar{f}_r(t,y(t))) + \mathscr{J}_2(g_{i,r}(y(t_i^-))). \tag{2.11}$$

It follows from (2.11) that

$$\begin{aligned}
\mathrm{e}^{-\rho t}||(\mathscr{T}_\xi y)(t+\tau)|| &\leq k\,\mathrm{e}^{-(\alpha+\rho)t+(\alpha+\varepsilon)\tau}||\xi|| \\
&\quad + k\,\mathrm{e}^{-\rho t}\left[\frac{\mathrm{Lip}(\bar{f}_r)}{\alpha-\rho} + \frac{\mathrm{Lip}(\bar{f}_r)}{\alpha+\rho} + \mathrm{Lip}(g_{i,r})(R_2+R_3)\right]||y(t+\tau)|| \\
&\leq k||\xi|| + 5k\left\{\frac{L(2r)}{\alpha-\rho} + \frac{L(2r)}{\alpha+\rho} + L(2r)(R_2+R_3)\right\}||y(\cdot+\tau)||_{PCB^0} \\
&\leq \frac{r}{2} + \frac{r}{4} + \frac{r}{4} = r,
\end{aligned}$$

that is, $||(\mathscr{T}_\xi y)(\cdot+\tau)||_{PCB^0} \leq r$ and $P(\tau)(\mathscr{T}_\xi y_1)(\tau) = P(\tau)\xi = \xi$, hence $\mathscr{T}$ maps $\mathscr{K}_\tau(\xi,r)$ into itself. Moreover, for any $\xi_1, \xi_2 \in \mathscr{K}_\tau(\xi,r)$, we have

$$\begin{aligned}
||(\mathscr{T}_\xi y_1) - (\mathscr{T}_\xi y_2)||_{PCB} &\leq 5k\left\{\frac{L(2r)}{\alpha-\rho} + \frac{L(2r)}{\alpha+\rho} + L(2r)(R_2+R_3)\right\}||y_1-y_2||_{PCB} \\
&\leq \frac{1}{2}||y_1-y_2||_{PCB},
\end{aligned}$$

hence $\mathscr{T}_\xi$ is a contraction in $\mathscr{K}_\tau(\xi,r)$ and has a unique fixed point $\varphi_\tau(\xi)(t)$ in $\mathscr{K}_\tau(\xi,r)$, which is the unique solution of the integral equation (2.1r) in $\mathscr{K}_\tau(\xi,r)$.

Now, from (2.8) and (2.10), we see that $\varphi_\tau(\xi)(t)$ is Lipschitz continuous in $\xi$ in the norm $||\cdot||$ and $\mathrm{Lip}(\varphi_\tau(\cdot)) \leq \frac{k\,\mathrm{e}^{-\rho\tau}}{1-\mathscr{L}} \leq 2k\,\mathrm{e}^{-\rho\tau}$. By lemma 2.8, we see $\varphi_\tau(\xi)(t+\tau)$ is also Lipschitz continuous in the norm $||\cdot||_{PCB^0}$ and $\mathrm{Lip}(\varphi_\tau(\cdot)(t+\tau)) \leq 2k$.

Let

$$\mathscr{M}_{\mathrm{loc}} := \{\xi + h(\tau,\xi)|\xi \in \mathscr{R}P(\tau) \cap B(0,\bar{r})\},$$

where

$$\begin{aligned}
h(\tau,\xi) = &-\int_0^\infty W(\tau,s+\tau)Q(s+\tau)\bar{f}_r(s+\tau,\varphi_\tau(\xi)(s+\tau))\,\mathrm{d}s \\
&-\sum_{i=1}^\infty W(\tau,s_i+\tau)Q(s_i+\tau)g_{i,r}(\varphi_\tau(\xi)(t_i^-+\tau)). \tag{2.12}
\end{aligned}$$

Therefore,

$$||y(\tau)|| \le ||\xi|| + ||h(\tau, \xi)||$$

$$\le ||\xi|| + k \int_0^\infty e^{-\alpha s} \cdot 5L(2r) \cdot ||\varphi_\tau(\xi)(s + \tau)|| \, ds$$

$$+ \sum_{i=1}^\infty k e^{-\alpha s_i} \cdot 5L(2r) \cdot ||\varphi_\tau(\xi)(t_i^- + \tau)||$$

$$\le ||\xi|| + 5kL(2r) \int_0^\infty e^{-(\alpha - \rho)s} \cdot ||\varphi_\tau(\xi)(\cdot + \tau)||_{PCB^0} \, ds$$

$$+ \sum_{i=1}^\infty 5kL(2r) e^{-(\alpha - \rho)s_i} \cdot ||\varphi_\tau(\xi)(\cdot + \tau)||_{PCB^0}$$

$$\le \frac{r}{2} + \frac{5kL(2r)}{\alpha - \rho} \cdot r + 5kL(2r) \cdot R_1 \cdot r$$

$$< \frac{r}{2} + \frac{r}{4} + \frac{r}{4} = r,$$

that is, $\mathscr{M}_{loc} \subset B(0, r)$. It follows from theorem 2.1 that $\mathscr{M}_{loc}$ is a local stable manifold of (1.2).

Moreover, for any $\xi_1, \xi_2 \in \mathscr{R}P(\tau) \cap B(0, \bar{r})$, it is clear that

$$||h(\tau, \xi_1) - h(\tau, \xi_2)|| \le \frac{5kL(2r)}{\alpha - \rho} \cdot 2k ||\xi_1 - \xi_2|| + \frac{5kL(2r)}{\alpha - \rho} \cdot 2k ||\xi_1 - \xi_2|| \le \frac{1}{2} ||\xi_1 - \xi_2||,$$

that is, $h(\tau, \xi)$ is Lipschitz continuous with respect to $\xi$ in the norm $|| \cdot ||$ and $\mathrm{Lip}(h) \le \frac{1}{2}$. This proves result (1) of theorem 2.2.

**Step 2: prove (2).** To prove (2), we define

$$H_\tau(\xi) = \xi + h(\tau, \xi), \quad \xi \in \mathscr{R}P(\tau) \cap B(0, \bar{r}).$$

Obviously, $H_\tau : \mathscr{R}P(\tau) \cap B(0, \bar{r}) \to \mathscr{M}_{loc}$ is continuous and $H_\tau(0) \equiv 0$. Since for any $\xi_1, \xi_2 \in \mathscr{R}P(\tau) \cap B(0, \bar{r})$, we obtain

$$||H_\tau(\xi_1) - H_\tau(\xi_2)|| \ge ||\xi_1 - \xi_2|| - \mathrm{Lip}(h)||\xi_1 - \xi_2|| \ge \frac{1}{2} ||\xi_1 - \xi_2||.$$

$H_\tau$ is one to one. Moreover, $H_\tau^{-1} = P(\tau)$ is a continuous mapping. Hence $H_\tau$ is a local homeomorphism.

**Step 3: prove (3).** To prove result (3), for any $\xi \in \mathscr{R}P(\tau) \cap B(0, \bar{r})$ define

$$L_r(\xi)(s) := L\left(\sigma\left(\frac{\varphi_\tau(\xi)(s)}{r}\right) ||\varphi_\tau(\xi)(s)||\right).$$

Clearly,

$$||h(\tau, \xi)|| \le k \int_0^\infty e^{-\alpha s} \cdot 5L_r(\xi)(s + \tau) ||\varphi_\tau(\xi)(s + \tau)|| \, ds$$

$$+ k \sum_{i=1}^\infty e^{-\alpha s_i} \cdot 5L_r(\xi)(t_i^- + \tau) ||\varphi_\tau(\xi)(t_i^- + \tau)||$$

$$\le 5k \cdot \mathrm{Lip}(\varphi_\tau(\xi)(\cdot + \tau)) ||\xi|| \cdot \int_0^\infty e^{(\rho - \alpha)s} L_r(\xi)(s + \tau) \, ds$$

$$+ 5k \cdot \mathrm{Lip}(\varphi_\tau(\xi)(\cdot + \tau)) ||\xi|| \cdot \sum_{i=1}^\infty e^{(\rho - \alpha)s_i} L_r(\xi)(t_i^- + \tau)$$

$$\le 10k^2 (\mathcal{D}_1(\xi) + \mathcal{D}_2(\xi)) ||\xi||,$$

where $\mathcal{D}_1(\xi) := \int_0^\infty e^{(\rho - \alpha)s} L_r(\xi)(s + \tau) \, ds$ and $\mathcal{D}_2(\xi) := \sum_{i=1}^\infty e^{(\rho - \alpha)s_i} L_r(\xi)(t_i^- + \tau)$.

Since $L_r(\xi) \le L(2r) < \infty$ and $\lim_{||\xi|| \to 0} \varphi_\tau(\xi) = 0$ uniformly in $\xi$, $\lim_{||\xi|| \to 0} \mathcal{D}_i(\xi) = 0$, $i = 1, 2$. Therefore,

$$\lim_{||\xi|| \to 0} \frac{||h(\tau, \xi) - h(\tau, 0)||}{||\xi|| - 0} = \lim_{||\xi|| \to 0} (\mathcal{D}_1(\xi) + \mathcal{D}_2(\xi)) = 0,$$

that is, $\mathcal{M}_{\text{loc}}$ is tangent to $\mathscr{R}P(\tau)$ at $\xi = 0$. ∎

## 3. Smoothness of stable manifolds

We give some notations as follows. Let $E_1, E_2$ be Banach space and $U$ be an open subset of $E_1$. For any integer $k \ge 0$, let

$$PC_\rho^k(\mathbb{I}, X) := \{x : \mathbb{I} \to X | x \in C^k((t_l, t_{l+1}], X), \quad \text{and } x^k(t_l^+), x^k(t_l^-) \text{ are well defined for each } l \in \mathbb{N}^+\},$$

and $||x||_{k,PCB} := ||x||_k \, e^{-\rho t}$, where $|| \cdot ||_k$ is the $k$th derivative of $x(t)$. Let

$$C^k(U, E_2) := \{f | f : U \to E_2 \text{ is } k \text{ times differentiable and } \sum_{i=0}^{k} \sup_{x \in U} |D^i f(x)| < \infty \text{ for } 0 \le i \le k.\}$$

and $|f|_k := \sum_{i=0}^{k} \sup_{x \in U} |D^i f(x)| < \infty$, where $D^i$ is the $i$th differentiation operator. Also let

$$C^{k,1}(U, E_2) := \left\{ f | f \in C^k(U, E_2) \text{ and } \sup_{x \ne y, \in U} \frac{|D^k f(x) - D^k f(y)|}{|x - y|} < \infty. \right\}$$

and $|f|_{k,1} := |f|_k + Lip D^k f$, where $Lip D^k f = \sup_{x \ne y, \in U} (|D^k f(x) - D^k f(y)|/|x - y|)$.

Clearly, $PC_\rho^k(\mathbb{I}, X), C^k(U, E_2), C^{k,1}(U, E_2)$ are all Banach space with norm $|| \cdot ||_{k,PCB}, |\cdot|_k, |\cdot|_{k,1}$, respectively. Let $L^k(E_1, E_2)$ be the Banach space of $k$ multilinear continuous maps from $E_1$ into $E_2$. For $\lambda \in L^k(E_1, E_2)$, $||\lambda||_k$ denotes the norm of $\lambda$.

The smoothness outside the jumping times of stable manifolds with respect to the time variable $\tau$ can be derived from the differentiability of solutions of the system. Therefore, it suffices to discuss the smoothness outside the jumping times of stable manifolds with respect to the space variable $y(\tau) \in X$.

**Theorem 3.1.** *Consider equation (1.2) with assumption $(H_1)$ and condition (2.1) in Banach space X. For any integer $k \ge 1$, assume that*

$(H_2^k)$ *for any $u \in X$, $\bar{f}(t, u) \in C^k(X, X)$, $\sup_{t \in \mathbb{R}_+} |\bar{f}(t, u)|_k < \infty$, $g_i(u) \in C^k(X, X)$, $\sup_{t \in t_i^-} |g_i(u(t_i^-))|_k < \infty$. Moreover, $\bar{f}(t, 0) \equiv 0, g_i(0) \equiv 0$.*

*Then equation (1.2) has a global $C^k$ stable manifold $\mathcal{M}$, which is given in theorem 2.1.*

**Theorem 3.2.** *Consider equation (1.2) with assumption $(H_1)$ in Hilbert space X. Assume that*

$(H_2^k)'$ *$\bar{f}(t, u), g_i(u)$ are continuously differentiable up to $k$th order in $u$, and*

$$||D_u^j \bar{f}(t, u)|| \le L_j(||u||) \quad \text{and} \quad ||D_u^j \bar{g}_i(u)|| \le L_j(||u||), \quad j = 1, 2, \ldots, k$$

*uniformly w.r.t. $t \in \mathbb{R}_+$, where $L_j(\cdot) : \mathbb{R}_+ \to \mathbb{R}_+$ is continuous and non-decreasing, $L_1(0) \equiv 0$, $\bar{f}(t, 0) \equiv 0, g_i(0) \equiv 0$.*

*Then equation (1.2) has a local stable manifold $\mathcal{M}_{\text{loc}}$, which is given in theorem 2.2 and is $C^k$ in u.*

In what follows, we only prove theorem 3.1. In fact, for a given $\sigma > 0$ small enough we consider the modified equation $(1.2r)$, we see that the assumption $(H_2^k)'$ implies that the modified nonlinear terms $\bar{f}_r(t, u), g_{i,r}(u)$ satisfy $(H_2^k)$. Hence, we can prove theorem 3.2 by theorem 3.1. It follows from lemma 2.7 that the solution $\varphi_\tau(\xi)(t) \in PC_\rho(\mathbb{I}, X)$ determines the structure of $\mathcal{M}$. Therefore, the aim is to prove the $C^k$ smoothness outside the jumping times of $\mathcal{M}$, we only need to prove the $C^k$

smoothness outside the jumping times of $\varphi_\tau(\xi)(t)$. Hence, the proof of theorem 3.1 is trivial if the following lemma is obtained.

**Lemma 3.3.** *Assume that all conditions of theorem 3.1 hold. Then the unique solution $\varphi_\tau(\xi)(t)$ of integral equation (1.3) in $PC_\rho(\mathbb{I}, X)$ is $C^k$ as a mapping $\varphi_\tau(\cdot) : \mathscr{R}P(\tau) \to PC_\rho^k(\mathbb{I}, X)$.*

We split the proof of lemma 3.3 into several lemmas. In what follows, we always assume that the assumptions in theorem 3.1 are satisfied.

Firstly, to prove lemma 3.3, we need a lemma from Hirsch *et al.* [5].

**Lemma 3.4 (Fibre Contraction Theorem).** *Let $E_1, E_2$ be Banach space and $V \subset E_1$ be a closed subset. Assume that (1) $\mathscr{B} : V \to V$, (2) $\mathscr{D}_x : E_2 \to E_2, x \in V$ and (3) $\mathscr{A}(x, y) = (\mathscr{B}(x), \mathscr{D}_x(y)), x \in V, y \in E_2$ are continuous maps. If $\mathscr{B}$ is a contraction and $\sup\{Lip(\mathscr{D}_x) | x \in V\} < 1$. Let the unique fixed point of $\mathscr{B}$ be $u$ and the unique fixed of $\mathscr{D}_u$ be $v$. Then $(u, v)$ is attractive, that is for any $(x, y) \in V \times E_2, \mathscr{A}^n(x, y) \to (u, v)$, as $n \to \infty$.*

For any $\psi(\xi) \in C^i(\mathscr{R}P(\tau), PC_\rho^i(\mathbb{I}, X))$ for $i = 1, 2, \ldots, k$, we define

$$\mathscr{F}_1^*(\psi)(t, \xi) := \int_\tau^t W(t, s) P(s) \cdot \bar{f}(s, \psi(\xi)(s)) \, ds,$$

$$\mathscr{F}_2^*(\psi)(t, \xi) := \int_t^\infty W(t, s) Q(s) \cdot \bar{f}(s, \psi(\xi)(s)) \, ds,$$

$$\mathscr{G}_1^*(\psi)(t_i^-, \xi) := \sum_{i=1}^{r(t, \tau)} W(t, s_i) P(s_i) \cdot g_i(\psi(\xi)(t_i^-))$$

and

$$\mathscr{G}_2^*(\psi)(t_i^-, \xi) := \sum_{i=r(t, \tau)+1}^\infty W(t, s_i) Q(s_i) \cdot g_i(\psi(\xi)(t_i^-)).$$

**Lemma 3.5.** *If $\psi \in C^i(\mathscr{R}P(\tau), PC_\rho^i(\mathbb{I}, X))$, then $\mathscr{F}_1^*, \mathscr{F}_2^*, \mathscr{G}_1^*, \mathscr{G}_2^* : \mathscr{R}P(\tau) \to PC_\rho^i(\mathbb{I}, X)$ are all $C^i$, for $i = 1, 2, \ldots, k$.*

*Proof.* For simplicity, we only prove $\mathscr{F}_1^*$ and $\mathscr{G}_1^*$, the others are similar to $\mathscr{F}_1^*$ and $\mathscr{G}_1^*$. We divide the proof of lemma 3.5 into a few claims. ∎

**Claim 3.6.** *If $\psi \in C^1(\mathscr{R}P(\tau), PC_\rho(\mathbb{I}, X))$, then $\mathscr{F}_1^* \in PC_\rho^1(\mathbb{I}, X)$.*

*Proof of claim 3.6.* Let $\psi(\xi) \in C^1(\mathscr{R}P(\tau), PC_\rho(\mathbb{I}, X))$ be fixed. If $\psi(\xi)$ is $C^1$ with respect to $\xi$, then we define a linear operator as follows:

$$\mathscr{D}_1^1(\psi)(t, \xi) \cdot \theta := \int_\tau^t W(t, s) \, P(s) \cdot D_x^1 \bar{f}(s, \psi(\xi)(s)) \cdot D\psi(\xi)(s)\theta \, ds, \tag{3.1}$$

where $\theta \in \mathscr{R}P(\tau)$ and $D\psi(\xi)(t)$ is the derivative of $\psi(\xi)(t)$ with respect to $\xi$. Since $\sup_{t \in \mathbb{R}_+} |f(t, \cdot)|_k < \infty$ and $\psi(\cdot)(t) \in C^1(\mathscr{R}P(\tau), PC_\rho(\mathbb{I}, X))$, we see that $\bar{f}(s, \psi(\cdot)(s)) \in C^1(\mathscr{R}P(\tau), PC_\rho(\mathbb{I}, X))$, that is, for any $\xi_1, \xi_2 \in \mathscr{R}P(\tau)$,

$$||\bar{f}(s, \psi(\xi_1)(s)) - \bar{f}(s, \psi(\xi_2)(s)) - D_x^1 \bar{f}(s, \psi(\xi_1)(s)) \cdot D\psi(\xi_1)(s)(\xi_1 - \xi_2)||_{1, PCB} = o(||\xi_1 - \xi_2||).$$

Hence,

$$\mathscr{K}_1 := ||\mathrm{e}^{-\rho t} \cdot \{\mathscr{F}_1^*(\psi)(t,\xi_1) - \mathscr{F}_1^*(\psi)(t,\xi_2) - \mathscr{D}_1^1(\psi)(t,\xi_1)(\xi_1 - \xi_2)\}||$$

$$= ||\mathrm{e}^{-\rho t} \cdot \int_\tau^t W(t,s)P(s)\{\bar{f}(s,\psi(\xi_1)(s)) - \bar{f}(s,\psi(\xi_2)(s))$$

$$- D_x^1 \bar{f}(s,\psi(\xi_1)(s)) \cdot D\psi(\xi_1)(s)(\xi_1 - \xi_2)\}\mathrm{d}s||$$

$$\le k\,\mathrm{e}^{-\rho t} \int_\tau^t \mathrm{e}^{-\alpha(t-s)} \cdot ||\bar{f}(s,\psi(\xi_1)(s)) - \bar{f}(s,\psi(\xi_2)(s))$$

$$- D_x^1 \bar{f}(s,\psi(\xi_1)(s)) \cdot D\psi(\xi_1)(s)(\xi_1 - \xi_2)||\,\mathrm{d}s$$

$$\le \frac{k}{\alpha + \rho} \cdot o(||\xi_1 - \xi_2||) = o(||\xi_1 - \xi_2||),$$

that is

$$||\mathscr{F}_1^*(\psi)(t,\xi_1) - \mathscr{F}_1^*(\psi)(t,\xi_2) - \mathscr{D}_1^1(\psi)(t,\xi_1)(\xi_1 - \xi_2)\}||_{1,PCB} = o(||\xi_1 - \xi_2||).$$

This shows $C^1$ smoothness of $\mathscr{F}_1^*$. ∎

**Claim 3.7.** If $\psi \in C^1(\mathscr{R}P(\tau), PC_\rho(\mathbb{I}, X))$, then $\mathscr{G}_1^* \in PC_\rho^1(\mathbb{I}, X)$.

*Proof of claim 3.7.* Similarly, we define

$$\mathscr{D}_2^1(\psi)(t_i^-, \xi) \cdot \theta := \sum_{i=1}^{r(t,\tau)} W(t,s_i)P(s_i)D_x^1 g_i(\psi(\xi)(t_i^-)) \cdot D\psi(\xi)(t_i^-) \cdot \theta. \tag{3.2}$$

Since $\sup_{t \in \mathbb{R}_+} |g_i(\cdot)|_k < \infty$ and $\psi(\cdot)(t) \in C^1$, we see $g_i \in C^1$. Hence,

$$\mathscr{K}_2 := ||\mathrm{e}^{-\rho t_i^-} \cdot \{\mathscr{G}_1^*(\psi)(t_i^-, \xi_1) - \mathscr{G}_1^*(\psi)(t_i^-, \xi_2) - \mathscr{D}_2^1(\psi)(t_i^-, \xi_1)(\xi_1 - \xi_2)\}||$$

$$\le k\,\mathrm{e}^{-\rho t_i^-} \cdot \sum_{i=1}^{r(t,\tau)} \mathrm{e}^{-\alpha(t-s_i)} \cdot ||g_i(\psi(\xi_1)(t_i^-)) - g_i(\psi(\xi_1)(t_i^-))$$

$$- D_x^1 g_i(\psi(\xi)(t_i^-)) \cdot D\psi(\xi)(t_i^-)(\xi_1 - \xi_2)||$$

$$\le kR_2 \cdot o(||\xi_1 - \xi_2||) = o(||\xi_1 - \xi_2||).$$

This shows that $\mathscr{G}_1^*$ is $C^1$ smoothness. ∎

**Claim 3.8.** If $\psi \in C^i(\mathscr{R}P(\tau), PC_\rho(\mathbb{I}, X))$, then $\mathscr{F}_1^* \in PC_\rho^i(\mathbb{I}, X)$ for $i = 1, 2, \ldots, k$.

*Proof of claim 3.8.* We will prove this claim by induction. Suppose that for $i = 1, 2, \ldots, k-1$, $\mathscr{F}_1^* \in PC_\rho^{i-1}(\mathbb{I}, X)$ is true. Computing formally $\mathscr{D}_1^2(\psi)(t,\xi)$, we see that $\mathscr{D}_1^2(\psi)(t,\xi)$ is determined by $D_x^2 \bar{f}(s, \psi(\xi)(s))$. That is

$$D_x^2 \bar{f}(s, \psi(\xi)(s)) = D_x^2 \bar{f}(s, \psi(\xi)(s)) \cdot (D\psi(\xi)(s))^2 + D_x \bar{f}(s, \psi(\xi)(s)) \cdot D^2\psi(\xi)(s).$$

Since $\sup_{t \in I} \mathrm{e}^{-\rho t} ||D\psi(\xi)(t)||_1 < \infty$ and $\sup_{t \in I} \mathrm{e}^{-2\rho t} ||D^2\psi(\xi)(t)||_2 < \infty$, we obtain

$$||D_x^2 \bar{f}(t, \psi(\xi)(t))||_{2,PCB} = \sup_{t \in I} \mathrm{e}^{-2\rho t} ||D_x^2 \bar{f}(t, \psi(\xi)(t))||_2 < \infty.$$

Hence, $\mathscr{D}_1^2(\psi)(t,\xi)$ has the same integral form as $\mathscr{D}_1^1(\psi)(t,\xi)$. Thus, it is not hard to see that $\mathscr{D}_1^{k-1}$ has the same integral form as $\mathscr{D}_1^1$. Using the same argument, we have $\mathscr{D}_1^{k-1}(\psi)(t,\xi)$ is $C^1$ from $\mathscr{R}P(\tau)$ to $PC_\rho^k(\mathbb{I}, X)$. ∎

**Claim 3.9.** If $\psi \in C^i(\mathscr{R}P(\tau), PC_\rho(\mathbb{I}, X))$, then $\mathscr{G}_1^* \in PC_\rho^i(\mathbb{I}, X)$ for $i = 1, 2, \ldots, k$.

*Proof of claim 3.9.* Claim 3.9 is similar to claim 3.8 and the proof is omitted. Finally, we complete the proof of lemma 3.5. ∎

Now we are going to prove lemma 3.3.

*Proof of lemma 3.3.* Lemma 2.7 implies that (1.3) has a unique solution $\varphi_\tau(\xi)(t) \in PC_\rho(\mathbb{I}, X)$ and $\varphi_\tau(\cdot): \mathscr{R}P(\tau) \to PC_\rho(\mathbb{I}, X)$ is $C^{0,1}$.

**Step 1.** To prove that $\varphi_\tau(\xi)(t) \in C^1(\mathscr{R}P(\tau), PC_\rho(\mathbb{I}, X))$. It suffices to show that $\psi(\cdot)(t) \in C^1(B, PC_\rho(\mathbb{I}, X))$ for any fixed but arbitrary bounded ball $B$ in $\mathscr{R}P(\tau)$, since the differentiability is a local property.

Let $E_1 = C^0(B, PC_\rho(\mathbb{I}, X))$ and $E_2 = C^0(B, \mathscr{L}^1(\mathscr{R}P(\tau), PC_\rho(\mathbb{I}, X)))$. Let $\psi \in E_1$, $\Psi \in E_2$, we define

$$\mathscr{B}(\psi)(t, \xi) = W(t, \tau)P(\tau)\xi + \mathscr{J}_1(\mathscr{F}(\psi(\xi)))(t) + \mathscr{J}_2(\mathscr{G}(\psi(\xi)))(t_k^-)$$

and

$$\begin{aligned}
\mathscr{D}_\psi(\Psi)(t, \xi) = {}& W(t, \tau)P(\tau) + \int_\tau^t W(t, s)P(s) \cdot D_x\bar{f}(s, \psi(\xi)(s)) \cdot \Psi \, ds \\
& - \int_t^\infty W(t, s)Q(s) \cdot D_x\bar{f}(s, \psi(\xi)(s)) \cdot \Psi \, ds \\
& + \sum_{i=1}^{r(t,\tau)} W(t, s_i)P(s_i) \cdot D_x g_i(\psi(\xi)(t_i^-)) \cdot \Psi \\
& - \sum_{i=r(t,\tau)+1}^{\infty} W(t, s_i)Q(s_i) \cdot D_x g_i(\psi(\xi)(t_i^-)) \cdot \Psi,
\end{aligned}$$

here $\mathscr{J}_1, \mathscr{J}_2, \mathscr{F}, \mathscr{G}$ are defined in lemma 2.7. By lemma 2.7, $\mathscr{B}$ is a contraction in $E_1$, let $\varphi_\tau(\xi)$ be the unique fixed point of $\mathscr{B}$. On the other hand, in the definition of $\mathscr{D}_\psi$, we suppose that for each $\theta \in \mathscr{R}P(\tau)$, $\mathscr{D}_\psi(\Psi)(t, \xi) \cdot \theta \in C^0(B, PC_\rho(\mathbb{I}, X))$ and is given by

$$\begin{aligned}
\mathscr{D}_\psi(\Psi)(t, \xi)\theta = {}& W(t, \tau)P(\tau)\theta + \int_\tau^t W(t, s)P(s) \cdot D_x\bar{f}(s, \psi(\xi)(s)) \cdot \Psi(\xi)(s) \cdot \theta \, ds \\
& - \int_t^\infty W(t, s)Q(s) \cdot D_x\bar{f}(s, \psi(\xi)(s)) \cdot \Psi(\xi)(s) \cdot \theta \, ds \\
& + \sum_{i=1}^{r(t,\tau)} W(t, s_i)P(s_i) \cdot D_x g_i(\psi(\xi)(t_i^-)) \cdot \Psi(\xi)(t_i^-) \cdot \theta \\
& - \sum_{i=r(t,\tau)+1}^{\infty} W(t, s_i)Q(s_i) \cdot D_x g_i(\psi(\xi)(t_i^-)) \cdot \Psi(\xi)(t_i^-) \cdot \theta.
\end{aligned}$$

Since $\mathcal{L} < 1$ and using the same argument as in (2.7), we obtain that

$$||\mathscr{D}_\psi(\Psi_1)(t, \xi)\theta - \mathscr{D}_\psi(\Psi_2)(t, \xi)\theta||_{1,PCB} \leq \mathcal{L}||\Psi_1(t, \xi)\theta - \Psi_2(t, \xi)\theta||_{1,PCB}.$$

Hence, $\mathscr{D}_\psi(\cdot)$ is a uniform contraction, and it has a unique fixed point $\Psi_\psi$ for each $\psi \in E_1$. Let $\Phi \in E_2$ be the unique fixed point of $\mathscr{D}_\psi(\cdot)$. We claim that $\Phi = D_\xi \varphi_\tau$. To prove this claim, let

$$\mathscr{A}(\psi, \Psi) = (\mathscr{B}(\psi), \mathscr{D}_\psi(\Psi)).$$

It follows from the Fibre Contraction Theorem that $(\varphi_\tau, \Phi)$ is an attractive fixed point of $\mathscr{A}$, that is, for every $\psi \in E_1$, $\Psi \in E_2$,

$$\mathscr{A}^n(\psi, \Psi) \to (\varphi_\tau, \Phi), \quad \text{as } n \to \infty,$$

where $\mathscr{A}^n$ denotes the $n$th iterate of $\mathscr{A}$. Moreover, fixed $\psi \in C^1(B, PC_\rho(\mathbb{I}, X))$, lemma 3.5 implies that $\mathscr{T}_\xi \psi \in C^1(B, PC_\rho(\mathbb{I}, X))$ and

$$
\begin{aligned}
D\mathscr{T}_\xi(\psi)(t, \xi) \cdot \theta = {} & W(t, \tau)P(\tau)\theta + \int_\tau^t W(t, s)P(s) \cdot D_x \bar{f}(s, \psi(\xi)(s))D\psi(\xi)(s) \cdot \theta \, ds \\
& - \int_t^\infty W(t, s)Q(s) \cdot D_x \bar{f}(s, \psi(\xi)(s))D\psi(\xi)(s) \cdot \theta \, ds \\
& + \sum_{i=1}^{r(t,\tau)} W(t, s_i)P(s_i) \cdot D_x g_i(\psi(\xi)(t_i^-))D\psi(\xi)(t_i^-) \cdot \theta \\
& - \sum_{i=r(t,\tau)+1}^\infty W(t, s_i)Q(s_i) \cdot D_x g_i(\psi(\xi)(t_i^-))D\psi(\xi)(t_i^-) \cdot \theta,
\end{aligned}
$$

for any $\theta \in \mathscr{R}P(\tau)$, where $\mathscr{T}_\xi$ is defined in (2.7). This means that $D\mathscr{T}_\xi(\psi) \in E_2$. Thus,

$$
\mathscr{A}(\psi, D\psi) = (\mathscr{B}(\psi), \mathscr{D}_\psi(D\psi)) = (\mathscr{B}(\psi), D\mathscr{B}(\psi)),
$$

for any $\psi \in C^1(B, PC_\rho(\mathbb{I}, X))$, from lemma 3.5, we see by induction that

$$
\mathscr{A}^2(\psi, D\psi) = (\mathscr{B}^2(\psi), \mathscr{D}_{\mathscr{B}(\psi)} \circ D\mathscr{B}(\psi)) = (\mathscr{B}^2(\psi), D\mathscr{B}^2(\psi))
$$

and

$$
\mathscr{A}^n(\psi, D\psi) = (\mathscr{B}^n(\psi), \mathscr{D}_{\mathscr{B}^{n-1}(\psi)} \circ \cdots \circ \mathscr{D}_{\mathscr{B}(\psi)} \circ D\mathscr{B}(\psi)) = (\mathscr{B}^n(\psi), D\mathscr{B}^n(\psi)).
$$

Note that

$$
\mathscr{D}_{\mathscr{B}^{n-1}(\psi)} \circ \cdots \circ \mathscr{D}_{\mathscr{B}(\psi)} \circ D\mathscr{B}(\psi) \in E_2.
$$

Then it follows from the attractiveness of $(\varphi_\tau, \Psi)$ that $\mathscr{B}^n \to \varphi_\tau$, $D\mathscr{B}^n \to \Phi$ as $n \to \infty$. Therefore $D\varphi_\tau = \Phi$ and $\varphi_\tau \in C^1$.

**Step 2.** To prove that $\varphi_\tau$ is $C^k$.

We assume that the result is true up to $k - 1$ and use induction. From lemma 3.5, it implies that $D^i\varphi_\tau(\xi)(t) \in C^0(B, PC_\rho^i(\mathbb{I}, X))$ for $i = 1, \ldots, k - 1$. Let

$$
E_1^k = C^0(B, \mathscr{L}^{k-1}(\mathscr{R}P(\tau), PC_\rho(\mathbb{I}, X)))
$$

and

$$
E_2^k = C^0(B, \mathscr{L}^k(\mathscr{R}P(\tau), PC_\rho(\mathbb{I}, X))).
$$

By differentiating and $\mathscr{D}_\psi$ formally, we define for any $\omega \in E_1^k$ and $\Omega \in E_2^k$

$$
\begin{aligned}
\mathscr{B}_k(\omega) = {} & \int_\tau^t W(t, s)P(s) \cdot D_x \bar{f}(s, \psi(\xi)(s))\omega(\xi)(s) \, ds + R_+^{k-2} \\
& - \int_t^\infty W(t, s)Q(s) \cdot D_x \bar{f}(s, \psi(\xi)(s))\omega(\xi)(s) \, ds + R_-^{k-2} \\
& + \sum_{i=1}^{r(t,\tau)} W(t, s_i)P(s_i) \cdot D_x g_i(\psi(\xi)(t_i^-))\omega(\xi)(t_i^-) + R_{++}^{k-2} \\
& - \sum_{i=r(t,\tau)+1}^\infty W(t, s_i)Q(s_i) \cdot D_x g_i(\psi(\xi)(t_i^-))\omega(\xi)(t_i^-) + R_{--}^{k-2}
\end{aligned}
$$

and

$$\mathcal{D}_{k,\omega}(\Omega) = \int_\tau^t W(t,s)P(s) \cdot D_x \bar{f}(s,\psi(\xi)(s))\Omega(\xi)(s)\,\mathrm{d}s + \int_\tau^t W(t,s)P(s)D_x^2\bar{f}(s,\psi(\xi)(s))$$

$$\times\, [(k-1)D\varphi_\tau(\xi)(s)\omega(\xi)(s) + \omega(\xi)(s)D\varphi_\tau(\xi)(s)]\,\mathrm{d}s + R_\ominus^{k-2}$$

$$-\int_t^\infty W(t,s)Q(s) \cdot D_x\bar{f}(s,\psi(\xi)(s))\Omega(\xi)(s)\,\mathrm{d}s - \int_t^\infty W(t,s)Q(s)D_x^2\bar{f}(s,\psi(\xi)(s))$$

$$\times\, [(k-1)D\varphi_\tau(\xi)(s)\omega(\xi)(s) + \omega(\xi)(s)D\varphi_\tau(\xi)(s)]\,\mathrm{d}s + R_\oplus^{k-2}$$

$$+\sum_{i=1}^{r(t,\tau)} W(t,s_i)P(s_i)D_x g_i(\psi(\xi)(t_i^-))\Omega(\xi)(t_i^-) + \sum_{i=1}^{r(t,\tau)} W(t,s_i)P(s_i)D_x^2 g_i(\psi(\xi)(t_i^-))$$

$$\times\, [(k-1)D\varphi_\tau(\xi)(t_i^-)\omega(\xi)(t_i^-) + \omega(\xi)(t_i^-)D\varphi_\tau(\xi)(t_i^-)] + R_{\ominus\ominus}^{k-2}$$

$$-\sum_{i=r(t,\tau)+1}^\infty W(t,s_i)Q(s_i)D_x g_i(\psi(\xi)(t_i^-))\Omega(\xi)(t_i^-)$$

$$-\sum_{i=r(t,\tau)+1}^\infty W(t,s_i)Q(s_i)D_x^2 g_i(\psi(\xi)(t_i^-))$$

$$\times\, [(k-1)D\varphi_\tau(\xi)(t_i^-)\omega(\xi)(t_i^-) + \omega(\xi)(t_i^-)D\varphi_\tau(\xi)(t_i^-)] + R_{\oplus\oplus}^{k-2},$$

where $R_+^{k-2}, R_-^{k-2}, R_{++}^{k-2}, R_{--}^{k-2}, R_\ominus^{k-2}, R_\oplus^{k-2}, R_{\ominus\ominus}^{k-2}, R_{\oplus\oplus}^{k-2}$ are appropriate terms involving derivatives of $\varphi_\tau$ with respect to $\xi$ of order at most $k-2$. Clearly, by the assumption of induction $D^{k-1}\varphi_\tau \in E_1^k$ is the unique fixed point of $\mathcal{B}$. For any $\Omega_1, \Omega_2 \in E_2^k$,

$$||\mathcal{D}_{k,\omega}(\Omega_1) - \mathcal{D}_{k,\omega}(\Omega_2)||_{k,PCB}$$

$$\leq k\,\mathrm{e}^{-k\rho t}\int_\tau^t \mathrm{e}^{-\alpha(t-s)} \cdot \mathrm{Lip}(\bar{f}) \cdot \mathrm{e}^{k\rho s}\mathrm{d}s ||\Omega_1 - \Omega_2||_{k,PCB}$$

$$+ k\,\mathrm{e}^{-k\rho t}\int_t^\infty \mathrm{e}^{\alpha(t-s)} \cdot \mathrm{Lip}(\bar{f}) \cdot \mathrm{e}^{k\rho s}\mathrm{d}s ||\Omega_1 - \Omega_2||_{k,PCB}$$

$$+ \sum_{k=1}^{r(t,\tau)} k\,\mathrm{e}^{-k\rho t_k^-}\,\mathrm{e}^{-\alpha(t-s_k)} \cdot \mathrm{Lip}(g_k) \cdot \mathrm{e}^{k\rho t_k^-} ||\Omega_1 - \Omega_2||_{k,PCB}$$

$$+ \sum_{k=r(t,\tau)+1}^\infty k\,\mathrm{e}^{-k\rho t_k^-}\,\mathrm{e}^{\alpha(t-s_k)} \cdot \mathrm{Lip}(g_k) \cdot \mathrm{e}^{k\rho t_k^-} ||\Omega_1 - \Omega_2||_{k,PCB}$$

$$\leq \mathcal{L} \cdot ||\Omega_1 - \Omega_2||_{k,PCB}.$$

This implies that $\mathcal{D}_{k,\omega}$ is uniform contraction. Using the same arguments as in Step 1, we see that $D^{k-1}\varphi_\tau : \mathcal{R}P(\tau) \to \mathcal{L}^{k-1}(\mathcal{R}P(\tau), PC_\rho(\mathbb{I}, X))$ is $C^1$. Therefore, $\varphi_\tau \in C^k$, we complete the proof of lemma 3.3. ∎

# 4. Periodicity of stable manifolds

Since the stable manifold of the non-instantaneous non-autonomous impulsive system (1.2) is dependent on the time variable $\tau$, its periodicity becomes very important.

**Theorem 4.1.** *Consider system (1.2) with assumptions $(H_1)$ and $(H_2')$ in Hilbert space $X$. Assume that*

$(A_1)$ $A(t+w) = A(t)$, for any $t \in \mathbb{T}$;

$(A_2)$ $t_{i+p} = t_i + w$ and $s_{i+p} = s_i + w$ for any $i \in \mathbb{N}^+$, where $p \in \mathbb{N}^+$ denotes the number of impulsive points and connection points of a periodic interval $[0,w]$ and set $s_0 = 0$, so $s_p = w$;

$(A_3)$ $B_{i+p}(t+w) = B_i(t)$, for any $i \in \mathbb{N}^+$ and $t \in \mathbb{J}$;

$(A_4)$ $\bar{f}(t + w, y) = \bar{f}(t, y)$, $P(t + w) = P(t)$ and $Q(t + w) = Q(t)$, for any $t \in \mathbb{R}_+$ and $y \in X$;

$(A_5)$ $g_{i+p}(\cdot) = g_i(\cdot)$ for any $i \in \mathbb{N}^+$, where $w$ is a given positive constant.

Then the local stable manifold $\mathcal{M}_{loc}$ of system (1.2) is periodic with period $w$, that is,

$$h(\tau + w, \xi) = h(\tau, \xi),$$

here $h$ is defined in theorem 2.2.

*Proof.* We firstly prove that the unique solution $\varphi_\tau(\xi)$ of the modified equation (1.2$r$) satisfies that

$$\varphi_{\tau+w}(\xi)(t + \tau + w) = \varphi_\tau(\xi)(t + \tau).$$

To prove this, we need the following statement.

**Lemma 4.2 ([43]).** *If $(A_1)$–$(A_3)$ hold, then $W(\cdot + w, \cdot + w) = W(\cdot, \cdot)$.*

We now proceed with our proof. It follows from the conditions $(A_1)$–$(A_5)$ and the proof of lemma 2.7 that

$$\varphi_\tau(\xi)(t) = W(t, \tau)P(\tau)\xi + \mathcal{J}_1(\mathcal{F}(\varphi_\tau(\xi)))(t) + \mathcal{J}_2(\mathcal{G}(\varphi_\tau(\xi)))(t),$$

where $\mathcal{F}, \mathcal{G}, \mathcal{J}_1, \mathcal{J}_2$ are defined in (2.6). Furthermore,

$$\varphi_{\tau+w}(\xi)(t + w) = W(t + w, \tau + w)P(\tau + w)\xi + \mathcal{J}_1(\mathcal{F}(\varphi_{\tau+w}(\xi)))(t + w) + \mathcal{J}_2(\mathcal{G}(\varphi_{\tau+w}(\xi)))(t + w),$$

where

$$
\begin{aligned}
\mathcal{J}_1(\mathcal{F}(\varphi_{\tau+w}(\xi)))(t + w) &= \int_{\tau+w}^{t+w} W(t + w, s)P(s)\bar{f}_r(s, \varphi_{\tau+w}(\xi)(s)) \, ds \\
&\quad - \int_{t+w}^{\infty} W(t + w, s)Q(s)\bar{f}_r(s, \varphi_{\tau+w}(\xi)(s)) \, ds \\
&= \int_{\tau}^{t} W(t + w, s + w)P(s + w)\bar{f}_r(s + w, \varphi_{\tau+w}(\xi)(s + w)) \, ds \\
&\quad - \int_{t}^{\infty} W(t + w, s + w)Q(s + w)\bar{f}_r(s + w, \varphi_{\tau+w}(\xi)(s + w)) \, ds \\
&= \int_{\tau}^{t} W(t, s)P(s)\bar{f}_r(s, \varphi_{\tau+w}(\xi)(s + w)) \, ds \\
&\quad - \int_{t}^{\infty} W(t, s)Q(s)\bar{f}_r(s, \varphi_{\tau+w}(\xi)(s + w)) \, ds,
\end{aligned}
$$

and similarly,

$$
\begin{aligned}
\mathcal{J}_2(\mathcal{G}(\varphi_{\tau+w}(\xi)))(t + w) &= \sum_{i=1}^{r(t+w,\tau+w)} W(t + w, s_i)P(s_i)g_{i,r}(\varphi_{\tau+w}(\xi)(t_i)) \\
&\quad - \sum_{i=r(t+w,\tau+w)+1}^{\infty} W(t + w, s_i)Q(s_i)g_{i,r}(\varphi_{\tau+w}(\xi)(t_i)) \\
&= \sum_{i=1}^{r(t,\tau)} W(t + w, s_i + w)P(s_i + w)g_{i+p,r}(\varphi_{\tau+w}(\xi)(t_i + w)) \\
&\quad - \sum_{i=r(t,\tau)+1}^{\infty} W(t + w, s_i + w)Q(s_i + w)g_{i+p,r}(\varphi_{\tau+w}(\xi)(t_i + w))
\end{aligned}
$$

$$= \sum_{i=1}^{r(t,\tau)} W(t,s_i)P(s_i)g_{i,r}(\varphi_{\tau+w}(\xi)(t_i+w))$$

$$- \sum_{i=r(t,\tau)+1}^{\infty} W(t,s_i)Q(s_i)g_{i,r}(\varphi_{\tau+w}(\xi)(t_i+w)).$$

Clearly,

$$P(\tau)\varphi_\tau(\xi)(t)\,|_{t=\tau} = \xi = P(\tau+w)\varphi_{\tau+w}(\xi)(t+w)\,|_{t=\tau},$$

thus $\varphi_\tau(\xi)(t)$ and $\varphi_{\tau+w}(\xi)(t+w)$ satisfy the same integral equation (1.2r) with the same initial value. By means of the uniqueness of the solution, we assert that

$$\varphi_\tau(\xi)(t) = \varphi_{\tau+w}(\xi)(t+w).$$

Now, it follows from (2.12) that

$$h(\tau+w,\xi) = -\int_0^\infty W(\tau+w,s+\tau+w)Q(s+\tau+w)\bar{f}_r(s+\tau+w,\varphi_{\tau+w}(\xi)(s+\tau+w))\,ds$$

$$- \sum_{i=1}^\infty W(\tau+w,s_i+\tau+w)Q(s_i+\tau+w)g_{i,r}(\varphi_{\tau+w}(\xi)(t_i+\tau+w))$$

$$= -\int_0^\infty W(\tau,s+\tau)Q(s+\tau)\bar{f}_r(s+\tau,\varphi_\tau(\xi)(s+\tau))\,ds$$

$$- \sum_{i=1}^\infty W(\tau,s_i+\tau)Q(s_i+\tau)g_{i,r}(\varphi_\tau(\xi)(t_i+\tau))$$

$$= h(\tau,\xi),$$

which implies the periodicity of $\mathcal{M}_{\text{loc}}$. Hence, we complete the proof of theorem 4.1. ∎

## 5. Duffing oscillators with non-instantaneous impulsive effects

A Duffing oscillator is an example of a periodically forced oscillator with a nonlinear elasticity. In this section, we consider the existence of stable manifold for the following nonlinear piecewise Duffing equations with non-instantaneous impulsive effects:

$$\begin{cases} \ddot{x} + \delta\dot{x} + \beta x + \alpha g(x) = \gamma\cos\omega t, & t \in \mathbb{T} := [s_{i-1}, t_i], \\ x(t_i^+) = ax(t_i^-) + \gamma\sin\omega t_i^-, & i \in \mathbb{N}^+, \\ \dot{x}(t_i^+) = b\dot{x}(t_i^-) + \gamma\cos\omega t_i^-, & i \in \mathbb{N}^+, \\ x(t) = ax(t_i^-) + \gamma\sin\omega t, & t \in \mathbb{J} := [t_i, s_i], \\ \dot{x}(t) = b\dot{x}(t_i^-) + \gamma\cos\omega t, & t \in \mathbb{J} := [t_i, s_i], \\ x(s_i^+) = x(s_i^-), & i \in \mathbb{N}^+, \\ \dot{x}(s_i^+) = \dot{x}(s_i^-), & i \in \mathbb{N}^+, \end{cases} \tag{5.1}$$

where $\delta > 0$ is a damping coefficient, $\beta, \alpha, \gamma$ are constant coefficients with physical significance, respectively, $a, b$ are positive constants, $\omega$ is a periodic coefficient (suppose that $\omega \le \min_{i\in\mathbb{N}^+}\{(t_i - s_{i-1}), (s_i - t_i)\}$). In addition, $g(x)$ is given by

$$g(x) = \begin{cases} 3x - 2, & x > 1; \\ x^3, & -1 \le x \le 1; \\ 3x + 2, & x < -1, \end{cases}$$

it is easy to obtain that for $x$ and $\bar{x}$,

$$|g(x) - g(\bar{x})| \le 3|x - \bar{x}|.$$

In equation (5.1), for simplicity, we take $\beta = -1, a = b$ and $\alpha, \gamma > 0$. For $\beta = -1 < 0$, the Duffing oscillator can be regarded as a system of a periodically forced steel beam that is deflected toward the two magnets, see [48].

Now, we are in a position to construct stable invariant manifold for equation (5.1). As usual, setting $x_1 = x, x_2 = \dot{x}$, we transform this problem into the following problem for the variable $(x_1, x_2)^T$:

$$
\begin{cases}
\begin{pmatrix} \dot{x}_1 \\ \dot{x}_2 \end{pmatrix} = \begin{pmatrix} 0 & 1 \\ 1 & -\delta \end{pmatrix} \begin{pmatrix} x_1 \\ x_2 \end{pmatrix} + \begin{pmatrix} 0 \\ -\alpha g(x_1) + \gamma \cos \omega t \end{pmatrix}, & t \in \mathbb{T}, \\[12pt]
\begin{pmatrix} x_1(t_i^+) \\ x_2(t_i^+) \end{pmatrix} = \begin{pmatrix} a & 0 \\ 0 & a \end{pmatrix} \begin{pmatrix} x_1(t_i^-) \\ x_2(t_i^-) \end{pmatrix} + \begin{pmatrix} \gamma \sin \omega t_i^- \\ \gamma \cos \omega t_i^- \end{pmatrix}, & i \in \mathbb{N}^+, \\[12pt]
\begin{pmatrix} x_1(t) \\ x_2(t) \end{pmatrix} = \begin{pmatrix} a & 0 \\ 0 & a \end{pmatrix} \begin{pmatrix} x_1(t_i^-) \\ x_2(t_i^-) \end{pmatrix} + \begin{pmatrix} \gamma \sin \omega t \\ \gamma \cos \omega t \end{pmatrix}, & t \in \mathbb{J}, \\[12pt]
\begin{pmatrix} x_1(s_i^+) \\ x_2(s_i^+) \end{pmatrix} = \begin{pmatrix} x_1(s_i^-) \\ x_2(s_i^-) \end{pmatrix}, & i \in \mathbb{N}^+.
\end{cases}
\tag{5.2}
$$

We see that there are two eigenvalues of matrix $\begin{pmatrix} 0 & 1 \\ 1 & -\delta \end{pmatrix}$: $\lambda_1 = -(\delta/2) - \left(\sqrt{\delta^2 + 4}/2\right)$ $(\lambda_1 < 0)$ and $\lambda_2 = -(\delta/2) + \left(\sqrt{\delta^2 + 4}/2\right)$ $(\lambda_2 > 0)$. Let $\Lambda = \begin{pmatrix} 1 & 1 \\ \lambda_1 & \lambda_2 \end{pmatrix}$, then $\Lambda^{-1} = \left(1/\sqrt{\delta^2 + 4}\right) \begin{pmatrix} \lambda_2 & -1 \\ -\lambda_1 & 1 \end{pmatrix}$.

Let $x = \Lambda y$, then equation (5.2) can be rewritten as

$$
\begin{cases}
\begin{pmatrix} \dot{y}_1 \\ \dot{y}_2 \end{pmatrix} = \begin{pmatrix} \lambda_1 & 0 \\ 0 & \lambda_2 \end{pmatrix} \begin{pmatrix} y_1 \\ y_2 \end{pmatrix} + \begin{pmatrix} \frac{-\lambda_1}{\sqrt{\delta^2+4}}[-\alpha g(w(y(t))) + \gamma \cos \omega t] \\ \frac{1}{\sqrt{\delta^2+4}}[-\alpha g(w(y(t))) + \gamma \cos \omega t] \end{pmatrix}, & t \in \mathbb{T}, \\[12pt]
\begin{pmatrix} y_1(t_i^+) \\ y_2(t_i^+) \end{pmatrix} = \begin{pmatrix} a & 0 \\ 0 & a \end{pmatrix} \begin{pmatrix} y_1(t_i^-) \\ y_2(t_i^-) \end{pmatrix} + \begin{pmatrix} \gamma \sin \omega t_i^- \\ \gamma \cos \omega t_i^- \end{pmatrix}, & i \in \mathbb{N}^+, \\[12pt]
\begin{pmatrix} y_1(t) \\ y_2(t) \end{pmatrix} = \begin{pmatrix} a & 0 \\ 0 & a \end{pmatrix} \begin{pmatrix} y_1(t_i^-) \\ y_2(t_i^-) \end{pmatrix} + \begin{pmatrix} \gamma \sin \omega t \\ \gamma \cos \omega t \end{pmatrix}, & t \in \mathbb{J}, \\[12pt]
\begin{pmatrix} y_1(s_i^+) \\ y_2(s_i^+) \end{pmatrix} = \begin{pmatrix} y_1(s_i^-) \\ y_2(s_i^-) \end{pmatrix}, & i \in \mathbb{N}^+,
\end{cases}
\tag{5.3}
$$

where $w(y(t)) = x_1(t) = y_1(t) + y_2(t)$. We see that the linear part of equation (5.3) admits an exponential dichotomy. In fact, if we take the initial value $y(\tau) = y_\tau$, $\tau \in (s_0, t_1)$, then for $s_i < t \le t_{i+1}$, the solution of the linear part of equation (5.3) is: $\tilde{y}(t) = W(t, \tau) y_\tau$, where $W(t, \tau)$ is the fundamental matrix. If we take $P(\tau) = \text{diag}\{1, 0\}, Q(\tau) = \text{diag}\{0, 1\}$, then

$$
|W(t, \tau) P(\tau)| = |e^{\lambda_1(t - s_i)} \Pi_{j=2}^{i} \{e^{\lambda_1(t_j - s_{j-1})} \cdot a\} e^{\lambda_1(t_1 - \tau)}| \le a\, e^{\lambda_1(t - \tau)}, \quad t \ge \tau
$$

and

$$
|W(t, \tau) Q(\tau)| = |e^{\lambda_2(t - s_i)} \Pi_{j=2}^{i} \{e^{\lambda_2(t_j - s_{j-1})} \cdot a\} e^{\lambda_2(t_1 - \tau)}| \le a\, e^{\lambda_2(t - \tau)}, \quad t \ge \tau.
$$

For $t_{i+1} < t \le s_{i+1}$, our results are similar. Hence, we show that the linear part of equation (5.3) admits an exponential dichotomy, i.e. the assumption $(H_1)$ in theorem 2.1 is satisfied. Moreover,

the assumption $(H_2)$ holds. Then, equation (5.3) has the following solution:

$$y(t) = W(t, \tau)P(\tau)y_\tau + \int_\tau^t W(t, s)P(s) \cdot \frac{-\lambda_1}{\sqrt{\delta^2 + 4}} \bar{g}(s, y(s)) \, ds$$

$$+ \sum_{i=1}^{r(t,\tau)} W(t, s_i)P(s_i) \cdot \gamma \sin \omega t_i^- - \sum_{i=r(t,\tau)+1}^{\infty} W(t, s_i)Q(s_i) \cdot \gamma \cos \omega t_i^-$$

$$- \int_t^\infty W(t, s)Q(s) \cdot \frac{1}{\sqrt{\delta^2 + 4}} \bar{g}(s, y(s)) \, ds, \tag{5.4}$$

where

$$\bar{g}(t, y(t)) = \begin{cases} -\alpha g(w(y(t))) + \gamma \cos \omega t, & t \in \mathbb{T}, \\ 0, & t \in \mathbb{J}. \end{cases}$$

Let $P(\tau)y_\tau = \xi$, if $\mathcal{L} := |3\alpha(1 + \lambda_2)/\lambda_2 \sqrt{\delta^2 + 4}| < 1$, then integral equation (5.4) has a unique solution $\varphi_\tau(\xi)(t)$ and for any $\xi_1, \xi_2 \in \mathcal{R}P(\tau)$,

$$|\varphi_\tau(\xi_1)(t) - \varphi_\tau(\xi_2)(t)| \le \frac{e^{\lambda_1(t-\tau)}}{1 - \mathcal{L}} |\xi_1 - \xi_2|, \quad \lambda_1 < 0, \ t \ge \tau.$$

We can now obtain the global Lipschitzian stable manifold $\mathcal{M}$ for equation (5.3). In fact, taking $t = \tau$ for equation (5.4), we have

$$y(\tau) = \varphi_\tau(\xi)(\tau) = \xi - \sum_{i=1}^\infty e^{\lambda_2(\tau - s_i)} \Pi_{j=2}^i \{e^{\lambda_2(s_{j-1} - t_j)} \cdot a\} \cdot \gamma \cos \omega t_i^-$$

$$- \int_\tau^\infty e^{\lambda_2(\tau - s_i)} \Pi_{j=2}^i \{e^{\lambda_2(s_{j-1} - t_j)} \cdot a\} e^{\lambda_2(s_i - s)} \cdot \frac{1}{\sqrt{\delta^2 + 4}} \bar{g}(s, y(s)) \, ds$$

$$:= \xi + h(\tau, \xi).$$

Clearly, $\mathcal{M} := \{\xi + h(\tau, \xi) | \xi \in \mathcal{R}P(\tau)\}$ is a stable manifold. We formulate this as a theorem.

**Theorem 5.1.** *The stable manifold of (5.1) is given by* $\mathcal{M} := \{\xi + h(\tau, \xi) | \xi \in \mathcal{R}P(\tau)\}$, *where*

$$h(\tau, \xi) = -\sum_{i=1}^\infty e^{\lambda_2(\tau - s_i)} \Pi_{j=2}^i \{e^{\lambda_2(s_{j-1} - t_j)} \cdot a\} \cdot \gamma \cos \omega t_i^-$$

$$- \int_\tau^\infty e^{\lambda_2(\tau - s_i)} \Pi_{j=2}^i \{e^{\lambda_2(s_{j-1} - t_j)} \cdot a\} e^{\lambda_2(s_i - s)} \cdot \frac{1}{\sqrt{\delta^2 + 4}} \bar{g}(s, y(s)) \, ds.$$

*Moreover, for any* $\xi_1, \xi_2 \in \mathcal{R}P(\tau)$,

$$|h(\tau, \xi_1) - h(\tau, \xi_2)| \le \frac{3|\alpha| \cdot |\xi_1 - \xi_2|}{(1 - \mathcal{L})\sqrt{\delta^2 + 4}}.$$

If the non-instantaneous impulsive effect reduces to instantaneous impulsive effect, i.e. the impulsive effect starts at an arbitrary point but it does not remain active on finite time intervals again. Thus system (5.1) reads as follows:

$$\begin{cases} \ddot{x} + \delta \dot{x} + \beta x + \alpha g(x) = \gamma \cos \omega t, & t \in \mathbb{R}^+ \setminus \mathbb{N}^+, \\ x(t_i^+) = ax(t_i^-) + \gamma \sin \omega t_i^-, & i \in \mathbb{N}^+, \\ \dot{x}(t_i^+) = a\dot{x}(t_i^-) + \gamma \cos \omega t_i^-, & i \in \mathbb{N}^+. \end{cases} \tag{5.5}$$

**Corollary 5.2.** *The Lipschitzian stable manifold of (5.5) is given by* $\mathcal{M} := \{\xi + h(\tau,\xi) | \xi \in \mathscr{R}P(\tau)\}$, *where*

$$h(\tau,\xi) = -\sum_{1\leq i\leq\infty} e^{\lambda_2(\tau-t_i)}\Pi^i_{j=2}\{e^{\lambda_2(t_{j-1}-t_j)}\cdot a\}\cdot\gamma\cos\omega t^-_i$$

$$-\int_\tau^\infty e^{\lambda_2(\tau-t_i)}\Pi^i_{j=2}\{e^{\lambda_2(t_{j-1}-t_j)}\cdot a\}\,e^{\lambda_2(t_i-s)}$$

$$\times\frac{1}{\sqrt{\delta^2+4}}[-\alpha g(w(\varphi_\tau(\xi)(s))) + \gamma\cos\omega s]\,\mathrm{d}s,\quad \tau\geq 0.$$

**Corollary 5.3.** *If the non-instantaneous impulsive effects are absent, then equation (5.1) reads as follows:*

$$\ddot{x} + \delta\dot{x} - x + \alpha g(x) = \gamma\cos\omega t,\quad t\geq 0. \tag{5.6}$$

*The Lipschitzian stable manifold of (5.1) is given by* $\mathcal{M} := \{\xi + h(\tau,\xi)|\xi\in\mathscr{R}P(\tau)\}$, *where*

$$h(\tau,\xi) = -\int_\tau^\infty e^{\lambda_2(\tau-s)}\cdot\frac{1}{\sqrt{\delta^2+4}}[-\alpha g(w(\varphi_\tau(\xi)(s))) + \gamma\cos\omega s]\,\mathrm{d}s,\quad \tau\geq 0.$$

*Proof.* Similar to the procedure just shown, we can immediately obtain the Lipschitzian stable manifold $\mathcal{M}$ for equations (5.5) and (5.6). ∎

**Example 5.4.** Simulation for the stable manifold of system (5.1).

Taking $\delta = \sqrt{5}, a = b = 1, \alpha = -1, \tau = 0$ and $\gamma = 1$ in system (5.1), we simulate the stable manifold by the successive approximation method as in the monograph of Perko [47]. We shall find the first two successive approximations $y^{(1)}(t,y)$ and $y^{(2)}(t,y)$, and use $y^{(2)}(t,y)$ to approximate the function $h_2$ describing the stable manifold

$$\mathcal{M}: y_2 = h_2(y_1),\quad y_1 = P(0)y,\ y_2 = Q(0)y.$$

We approximate the solution of the integral equation

$$y(t,y) = \begin{pmatrix} W(t,0)P(0)y \\ 0 \end{pmatrix} + \begin{pmatrix} \int_0^t W(t,s)P(s)\cdot\frac{-\lambda_1}{3}\bar{g}(s,y(s))\,\mathrm{d}s \\ -\int_t^\infty W(t,s)Q(s)\cdot\frac{1}{3}\bar{g}(s,y(s))\,\mathrm{d}s \end{pmatrix}$$

$$+\begin{pmatrix} \sum_{i=1}^{r(t,0)} W(t,s_i)P(s_i)\cdot\sin\omega t \\ -\sum_{i=r(t,0)+1}^\infty W(t,s_i)Q(s_i)\cdot\cos\omega t \end{pmatrix}.$$

By the successive approximations

$$y^0(t,y) = 0,$$

$$y^{(1)}(t,y) = \begin{pmatrix} W(t,0)P(0)y \\ 0 \end{pmatrix},$$

$$y^{(2)}(t,y) = \begin{pmatrix} W(t,0)P(0)y \\ 0 \end{pmatrix} + \begin{pmatrix} \int_0^t W(t,s)P(s)\cdot\frac{-\lambda_1}{3}\bar{g}(s,y^{(1)}(s,y))\,\mathrm{d}s \\ -\int_t^\infty W(t,s)Q(s)\cdot\frac{1}{3}\bar{g}(s,y^{(1)}(s,y))\,\mathrm{d}s \end{pmatrix}$$

$$+\begin{pmatrix} \sum_{i=1}^{r(t,0)} W(t,s_i)P(s_i)\cdot\sin\omega t \\ -\sum_{i=r(t,0)+1}^\infty W(t,s_i)Q(s_i)\cdot\cos\omega t \end{pmatrix}.$$

If we take $|y| \leq 1$, then

$$\bar{g}(s, y^{(1)}(s, y)) = g(s, y^{(1)}(s, y)) + \cos \omega t = (W(t, 0)P(0))^3 y_1^3 + \cos \omega t.$$

In view of

$$|W(t, 0)P(0)| \leq e^{\lambda_1 t} \quad \text{and} \quad |W(t, 0)Q(0)| \leq e^{\lambda_2 t}, \quad t \geq 0,$$

we assert that the approximate solutions do indeed limit to the origin as $t \to \infty$. To obtain a picture of the stable manifold, it is sufficient to plot the curve as a function of the initial point at any value of $t$, saying, for example, at $t = 0$ (figure 1). In this case, we have a parametric representation of the approximate stable manifold

$$\mathscr{M}_{\mathrm{loc}}(0, 0) \approx \{(y_1, h_2(y_1)) \mid h_2(y_1) = c y_1^3, |y_1| \leq 1, c > 0\}.$$

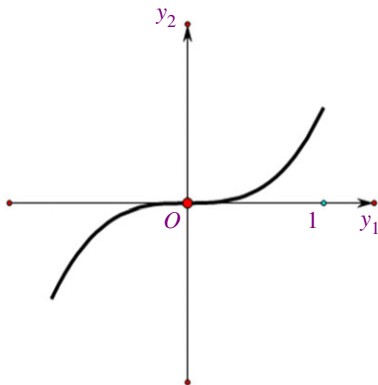

**Figure 1.** The approximate stable manifold. (Online version in colour.)

**Data accessibility.** This article has no additional data.

**Authors' contributions.** W.L.: conceptualization, data curation, formal analysis, investigation, writing—original draft; M.P.: formal analysis, funding acquisition, investigation, validation, writing—review and editing; Y.X.: formal analysis, funding acquisition, investigation, methodology, resources, supervision, writing—original draft, writing—review and editing.

All authors gave final approval for publication and agreed to be held accountable for the work performed therein.

**Competing interests.** We declare we have no competing interests.

**Funding.** This paper was jointly supported from the National Natural Science Foundation of China under grant nos (11931016, 11671176), grant Fondecyt no. (1170466) and Natural Science Foundation of Zhejiang Province under grant no. (LY20A010016).

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
