## [Peer Review File · Proceedings. Mathematical, Physical, and Engineering Sciences]

Review History

RSPA-2021-0957.R0 (Original submission)

Review form: Referee 1

Is the manuscript an original and important contribution to its field?

Good

Is the paper of sufficient general interest?

Good

Is the overall quality of the paper suitable?

Good

Can the paper be shortened without overall detriment to the main message?

Yes

Do you think some of the material would be more appropriate as an electronic appendix?

Yes

Do you have any ethical concerns with this paper?

No

Recommendation?

Accept with minor revision (please list in comments)

Comments to the Author(s)

The article contains new and correct results on the theory of differential equations with non-instantaneous impulses.

I suggest including a motivation on the invariant manifold theory role in dynamical systems.

What is the relation of the techniques used with variational methods?

Variational approach to differential equations with not instantaneous impulses. Bai, L., Nieto, J.J. Applied Mathematics Letters 73 (2017), pp. 44-48.

Some recent references on non-instantaneous impulsive equations and their applications:

On a delayed epidemic model with non-instantaneous impulses. Bai, L., Nieto, J.J., Uzal, J.M. Communications on Pure and Applied Analysis 19 (2020), pp. 1915-1930.

Abstract impulsive differential equations without predefined time impulses. Hernández, E. Journal of Mathematical Analysis and Applications 491 (2020), 124288.

And, of course, the monograph:

Agarwal, R., Hristova, S., O'Regan, D., Non-instantaneous impulses in differential equations. Springer, 2017, 251 pages.

Include some examples with graphs to illustrate the theory.

In the title it appears "equation in a Banach space", but section 4 "Duffing oscillators with non-instantaneous impulsive effects" seems one-dimensional.

Review form: Referee 2

Is the manuscript an original and important contribution to its field?

Excellent

Is the paper of sufficient general interest?

Excellent

Is the overall quality of the paper suitable?

Excellent

Can the paper be shortened without overall detriment to the main message?

Yes

Do you think some of the material would be more appropriate as an electronic appendix?

Yes

Do you have any ethical concerns with this paper?

No

Recommendation?

Accept with minor revision (please list in comments)

Comments to the Author(s)

Please see the attachment.

Decision letter (RSPA-2021-0957.R0)

31-Jan-2022

Dear Dr Xia,

On behalf of the Editor, I am pleased to inform you that your Manuscript RSPA-2021-0957 entitled "Smooth stable manifolds for non-instantaneous impulsive equations on Banach space with applications to Duffing oscillators" has been accepted for publication subject to minor revisions in Proceedings A. Please find the referees' comments below.

The reviewer(s) have recommended publication, but also suggest some minor revisions to your manuscript. Therefore, I invite you to respond to the reviewer(s)' comments and revise your manuscript. Please note that we have a strict upper limit of 28 pages for each paper. Please endeavour to incorporate any revisions while keeping the paper within journal limits. Please note that page charges are made on all papers longer than 20 pages. If you cannot pay these charges you must reduce your paper to 20 pages before submitting your revision. Your paper has been ESTIMATED to be 27 pages. We cannot proceed with typesetting your paper without your agreement to meet page charges in full should the paper exceed 20 pages when typeset. If you have any questions, please do get in touch.

It is a condition of publication that you submit the revised version of your manuscript within 7 days. If you do not think you will be able to meet this date please let me know in advance of the due date.

To revise your manuscript, log into <https://mc.manuscriptcentral.com/prsa> and enter your Author Centre, where you will find your manuscript title listed under "Manuscripts with Decisions." Under "Actions," click on "Create a Revision." Your manuscript number has been appended to denote a revision.

You will be unable to make your revisions on the originally submitted version of the manuscript. Instead, revise your manuscript and upload a new version through your Author Centre.

When submitting your revised manuscript, you will be able to respond to the comments made by the referee(s) and upload a file "Response to Referees" in Step 1: "View and Respond to Decision Letter". Please provide a point-by-point response to the comments raised by the reviewers and the editor(s). A thorough response to these points will help us to assess your revision quickly. You can also upload a 'tracked changes' version either as part of the 'Response to reviews' or as a 'Main document'.

IMPORTANT: Your original files are available to you when you upload your revised manuscript. Please delete any redundant files before completing the submission process.

When uploading your revised files, please make sure that you include the following as we cannot proceed without these:

- 1) A text file of the manuscript (doc, txt, rtf or tex), including the references, tables (including captions) and figure captions. Please remove any tracked changes from the text before submission. PDF files are not an accepted format for the "Main Document".
- 2) A separate electronic file of each figure (tif, eps or print-quality pdf preferred). The format should be produced directly from original creation package, or original software format.
- 3) Electronic Supplementary Material (ESM): all supplementary materials accompanying an accepted article will be treated as in their final form. Note that the Royal Society will not edit or typeset supplementary material and it will be hosted as provided. Please ensure that the supplementary material includes the paper details where possible (authors, article title, journal name). Supplementary files will be published alongside the paper on the journal website and posted on the online figshare repository (<https://figshare.com>). The heading and legend provided for each supplementary file during the submission process will be used to create the figshare page, so please ensure these are accurate and informative so that your files can be found in searches. Files on figshare will be made available approximately one week before the accompanying article so that the supplementary material can be attributed a unique DOI. Alternatively you may upload a zip folder containing all source files for your manuscript as described above with a PDF as your "Main Document". This should be the full paper as it appears when compiled from the individual files supplied in the zip folder.

Article Funder

Please ensure you fill in the Article Funder question on page 2 to ensure the correct data is collected for FundRef (<http://www.crossref.org/fundref/>).

Media summary

Please ensure you include a short non-technical summary (up to 100 words) of the key findings/importance of your paper. This will be used for to promote your work and marketing purposes (e.g. press releases). The summary should be prepared using the following guidelines:

- *Write simple English: this is intended for the general public. Please explain any essential technical terms in a short and simple manner.
- *Describe (a) the study (b) its key findings and (c) its implications.
- *State why this work is newsworthy, be concise and do not overstate (true 'breakthroughs' are a rarity).
- *Ensure that you include valid contact details for the lead author (institutional address, email address, telephone number).

Cover images

We welcome submissions of images for possible use on the cover of Proceedings A. Images should be square in dimension and please ensure that you obtain all relevant copyright permissions before submitting the image to us. If you would like to submit an image for consideration please send your image to proceedingsa@royalsociety.org

Open Access

You are invited to opt for open access, our author pays publishing model. Payment of open access fees will enable your article to be made freely available via the Royal Society website as soon as it is ready for publication. For more information about open access please visit <https://royalsociety.org/journals/authors/open-access/>. The open access fee for this journal is

£1700/\$2380/€2040 per article. VAT will be charged where applicable. Please note that if the corresponding author is at an institution that is part of a Read and Publishing deal you are required to select this option. See <https://royalsociety.org/journals/librarians/purchasing/read-and-publish/read-publish-agreements/> for further details.

Once again, thank you for submitting your manuscript to Proceedings A and I look forward to receiving your revision. If you have any questions at all, please do not hesitate to get in touch.

Best wishes
Raminder Shergill
proceedingsa@royalsociety.org
Proceedings A

on behalf of
Professor Fioralba Cakoni
Board Member
Proceedings A

Reviewer(s)' Comments to Author:

Referee: 1

Comments to the Author(s)

The article contains new and correct results on the theory of differential equations with non-instantaneous impulses.

I suggest including a motivation on the invariant manifold theory role in dynamical systems.

What is the relation of the techniques used with variational methods?

Variational approach to differential equations with not instantaneous impulses. Bai, L., Nieto, J.J. Applied Mathematics Letters 73 (2017), pp. 44-48.

Some recent references on non-instantaneous impulsive equations and their applications:

On a delayed epidemic model with non-instantaneous impulses. Bai, L., Nieto, J.J., Uzal, J.M. Communications on Pure and Applied Analysis 19 (2020), pp. 1915-1930.

Abstract impulsive differential equations without predefined time impulses. Hernández, E. Journal of Mathematical Analysis and Applications 491 (2020), 124288.

And, of course, the monograph:

Agarwal, R., Hristova, S., O'Regan, D., Non-instantaneous impulses in differential equations. Springer, 2017, 251 pages.

Include some examples with graphs to illustrate the theory.

In the title it appears "equation in a Banach space", but section 4 "Duffing oscillators with non-instantaneous impulsive effects" seems one-dimensional.

Referee: 2

Comments to the Author(s)

Please see the attachment.

Board member pre-assessment comments (if available):

The topic and the method of this paper are sufficiently interesting. The significance of the contribution is high and the paper deserves to undergo a full peer review.

Author's Response to Decision Letter for (RSPA-2021-0957.R0)

See Appendix A.

Decision letter (RSPA-2021-0957.R1)

07-Feb-2022

Dear Dr Xia

I am pleased to inform you that your manuscript entitled "Smooth stable manifolds for non-instantaneous impulsive equations with applications to Duffing oscillators" has been accepted in its final form for publication in Proceedings A.

Our Production Office will be in contact with you in due course. You can expect to receive a proof of your article soon. Please contact the office to let us know if you are likely to be away from e-mail in the near future. If you do not notify us and comments are not received within 5 days of sending the proof, we may publish the paper as it stands.

As a reminder, you have provided the following 'Data accessibility statement' (if applicable). Please remember to make any data sets live prior to publication, and update any links as needed when you receive a proof to check. It is good practice to also add data sets to your reference list. Statement (if applicable):

Open access

You are invited to opt for open access, our author pays publishing model. Payment of open access fees will enable your article to be made freely available via the Royal Society website as soon as it is ready for publication. For more information about open access please visit <https://royalsociety.org/journals/authors/which-journal/open-access/>. The open access fee for this journal is £1700/\$2380/€2040 per article. VAT will be charged where applicable.

Note that if you have opted for open access then payment will be required before the article is published – payment instructions will follow shortly.

If you wish to opt for open access then please inform the editorial office (proceedingsa@royalsociety.org) as soon as possible.

Your article has been estimated as being 27 pages long. Our Production Office will inform you of the exact length at the proof stage.

Proceedings A levies charges for articles which exceed 20 printed pages. (based upon approximately 540 words or 2 figures per page). Articles exceeding this limit will incur page charges of £150 per page or part page, plus VAT (where applicable).

Under the terms of our licence to publish you may post the author generated postprint (ie. your accepted version not the final typeset version) of your manuscript at any time and this can be made freely available. Postprints can be deposited on a personal or institutional website, or a recognised server/repository. Please note however, that the reporting of postprints is subject to a media embargo, and that the status the manuscript should be made clear. Upon publication of the definitive version on the publisher's site, full details and a link should be added.

You can cite the article in advance of publication using its DOI. The DOI will take the form: 10.1098/rspa.XXXX.YYYY, where XXXX and YYYY are the last 8 digits of your manuscript number (eg. if your manuscript number is RSPA-2017-1234 the DOI would be 10.1098/rspa.2017.1234).

For tips on promoting your accepted paper see our blog post:
<https://royalsociety.org/blog/2020/07/promoting-your-latest-paper-and-tracking-your-results/>

On behalf of the Editor of Proceedings A, we look forward to your continued contributions to the Journal.

Sincerely,
Raminder Shergill
proceedingsa@royalsociety.org

on behalf of
Professor Fioralba Cakoni
Board Member
Proceedings A

Appendix A

Ms. Ref. No.: RSPA-2021-0957

Title: Smooth stable manifolds for non-instantaneous impulsive equations on Banach space with applications to Duffing oscillators

Authors: Weijie Lu, Manuel Pinto, Yonghui Xia

Decision letter from Proceedings A

Dear Dr Xia,

On behalf of the Editor, I am pleased to inform you that your Manuscript RSPA-2021-0957 entitled “Smooth stable manifolds for non-instantaneous impulsive equations on Banach space with applications to Duffing oscillators” has been accepted for publication subject to minor revisions in Proceedings A. Please find the referees’ comments below.

Response to the Editors and all Reviewers:

Thank you very much for your time to review the manuscript, and provide valuable comments which improve the presentation of this paper. According to the reviewers’ suggestions, we have revised the manuscript point by point and marked in red in this revision. We response to the reviewers’ comments one by one as follows.

Comments from Reviewer 1: The article contains new and correct results on the theory of differential equations with non-instantaneous impulses.

I suggest including a motivation on the invariant manifold theory role in dynamical systems.

What is the relation of the techniques used with variational methods? Variational approach to differential equations with not instantaneous impulses. Bai, L., Nieto, J.J. Applied Mathematics Letters 73 (2017), pp. 44-48.

Some recent references on non-instantaneous impulsive equations and their applications:

On a delayed epidemic model with non-instantaneous impulses. Bai, L., Nieto, J.J., Uzal, J.M. Communications on Pure and Applied Analysis 19 (2020), pp. 1915-1930.

Abstract impulsive differential equations without predefined time impulses. Hernández, E. Journal of Mathematical Analysis and Applications 491 (2020), 124288.

And, of course, the monograph:

Agarwal, R., Hristova, S., O’Regan, D., Non-instantaneous impulses in differential

equations. Springer, 2017, 251 pages.

Include some examples with graphs to illustrate the theory.

In the title it appears “equation in a Banach space”, but section 4 “Duffing oscillators with non-instantaneous impulsive effects” seems one-dimensional.

Response to Reviewer 1’s comments:

Response: Thank the anonymous reviewer for his/her positive comments and valuable suggestions which improve the presentation of this paper.

1. I suggest including a motivation on the invariant manifold theory role in dynamical systems.

Response: Thank you very much for your good idea. We mentioned the role of the invariant manifold theory in the first paragraph in Introduction.

2. What is the relation of the techniques used with variational methods? Variational approach to differential equations with not instantaneous impulses. Bai, L., Nieto, J.J. Applied Mathematics Letters 73 (2017), pp. 44-48.

Response: We discussed it as you suggested.

3. Some recent references on non-instantaneous impulsive equations and their applications:

On a delayed epidemic model with non-instantaneous impulses. Bai, L., Nieto, J.J., Uzal, J.M. Communications on Pure and Applied Analysis 19 (2020), pp. 1915-1930.

Abstract impulsive differential equations without predefined time impulses. Hernández, E. Journal of Mathematical Analysis and Applications 491 (2020), 124288.

And, of course, the monograph:

Agarwal, R., Hristova, S., O’Regan, D., Non-instantaneous impulses in differential equations. Springer, 2017, 251 pages.

Response: Thank you very much for the recent references. We added all the recent references and discussed these in the introduction.

4. Include some examples with graphs to illustrate the theory.

Response: Thank you very much. Simulation with graph is added to illustrate the theory.

5. In the title it appears “equation in a Banach space”, but section 4 “Duffing

oscillators with non-instantaneous impulsive effects” seems one-dimensional.

Response: Our main purpose to give a simple, interesting but effective application is given to show the feasibility and effectiveness of our theory. For the sake of easily understanding, we choose Duffing oscillators with non-instantaneous impulsive effects as an application. This has no conflicts with Banach space. For the sake of shortening the title and avoiding confusion, we remove the word “in a Banach space” in the title. Thank you for kind reminder.

Comments from Reviewer 2: In this manuscript, the authors established a theory of smooth stable manifolds for the non-instantaneous impulsive differential equations on the Banach space. Furthermore, C^k -smoothness of these stable manifolds was obtained. Finally, an application to nonlinear Duffing oscillators with non-instantaneous impulsive effects is given, to demonstrate the existence of stable manifold. The approach is based on the method of Lyapunov-Perron and Hadamard. In fact, the theory of smooth stable manifolds for ODE is well known and very important. However, it is not easy to extend it to the non-instantaneous impulsive differential equation (for short, NIIDE). The NIIDE is a new kind of differential equation which has its complexity. The authors overcome this difficulty and successfully extended the classical theory of smooth stable manifolds to NIIDE. This is the first contribution of this paper.

The authors used a more general Banach space $(PC_\rho(\mathbb{I}, X), \|x\|_{PCB})$ with norms $\|x\|_{PCB}$ in this paper. Clearly, $(PC_\rho(\mathbb{I}, X), \|x\|_{PCB})$ is a more general Banach space than the space $(PC(\mathbb{I}, X), \|x\|)$. Consequently, they generalized and weakened the conditions of the known results on stable manifolds (e.g. see Henry [Lecture Notes in Mathematics, 1981], Chow and Lu [JDE 1988, Proc. Roy. Soc. Edinburgh. Sect. A, 1988], Bates and Jones [Dynamics. Rep. 1989], Bates et al. [Mem. Amer. Math. Soc., 1998] and so on). In this sense, this paper significantly improved some classical previous results. This is another highlight of the manuscript.

Base on the above contributions and the novelty, this paper is strongly recommended for publication in a good journal. However, before acceptance, a few points should be addressed.

1. The periodicity of the invariant manifold is also interesting. Is it possible to discuss the periodicity of the invariant manifolds for the non-instantaneous impulsive differential equation? If the authors can discuss this, it would be better.

2. If the non-instantaneous impulsive effects reduce to the instantaneous impulsive effects, or if the impulsive effects are absent, the obtained main results reduces to some special cases. I suggest to add some corollaries or remarks after Theorem 2.1 and 2.2 to address this point. It would be better with some comparison.

3. Last sentence of page 3, Gautam, Dabas [36] should be: Gautam and Dabas [36].

4. The first sentence in subsection 1.3 is not very clear. Please check it.

5. After first sentence of Step 1 in the proof of Lemma 3.3, please check the grammar “It is suffices”.

6. Before (1.2), “are”??

7. The sentence before Theorem 2.3, “Theorems”??

8. Please check the paragraph before Theorem 3.1 on page 17. The sentences are confusing. It is better to say: The smoothness outside the jumping times of stable manifolds with respect to the time variable τ can be derived from the differentiability of solutions of the system. Therefore, it suffices to discuss the smoothness outside the jumping times of stable manifolds with respect to the space variable $y(\tau) \in X$.

9. It is better to move the first sentence in the proof of Lemma 3.4 to the position before Lemma 3.4. You can mention there: To prove Lemma 3.3, we need a lemma from Hirsch et al. [5].

10. Replace “impulse equation” by “impulsive equation”. Please check it throughout the paper.

Response to Reviewer 2's comments:

Response: Thank the anonymous reviewer for his/her valuable suggestions which improve the presentation of this paper.

1. The periodicity of the invariant manifold is also interesting. Is it possible to discuss the periodicity of the invariant manifolds for the non-instantaneous impulsive differential equation? If the authors can discuss this, it would be better.

Response: Thank you, it is a good opinion, we added the periodicity of the stable manifold in Section 4.

2. If the non-instantaneous impulsive effects reduce to the instantaneous im-

pulsive effects, or if the impulsive effects are absent, the obtained main results reduces to some special cases. I suggest to add some corollaries or remarks after Theorem 2.1 and 2.2 to address this point. It would be better with some comparison.

Response: Yes, as you suggest, we added results on instantaneous impulsive effects and the disappearance of impulsive effects (see Corollary 2.3 and 2.4). Moreover, Remark 2.5 including the comparisons is given.

3. Last sentence of page 3, Gautam, Dabas [36] should be: Gautam and Dabas [36].

Response: We fixed it.

4. The first sentence in subsection 1.3 is not very clear. Please check it.

Response: Thank you. We modified the first sentence to make it clearer.

5. After first sentence of Step 1 in the proof of Lemma 3.3, please check the grammar “It is suffices”.

Response: We fixed it and removed the word “is”.

6. Before (1.2), “are”??

Response: We fixed it, “are” was replaced by “is”.

7. The sentence before Theorem 2.3, “Theorems”??

Response: We fixed it.

8. Please check the paragraph before Theorem 3.1 on page 17. The sentences are confusing. It is better to say: The smoothness outside the jumping times of stable manifolds with respect to the time variable τ can be derived from the differentiability of solutions of the system. Therefore, it suffices to discuss the smoothness outside the jumping times of stable manifolds with respect to the space variable $y(\tau) \in X$.

Response: Thank you very much, we fixed it as you suggested.

9. It is better to move the first sentence in the proof of Lemma 3.4 to the position before Lemma 3.4. You can mention there: To prove Lemma 3.3, we need a lemma from Hirsch et al. [5].

Response: Yes, we fixed it.

10. Replace “impulse equation” by “impulsive equation”. Please check it throughout the paper.

Response: We fixed it throughout the paper. Thank you.